# Integrative frontal-parietal dynamics supporting cognitive control

**Derek Evan Nee\***

Department of Psychology, Florida State University, Tallahassee, United States

**Abstract** Coordinating among the demands of the external environment and internal plans requires cognitive control supported by a fronto-parietal control network (FPCN). Evidence suggests that multiple control systems span the FPCN whose operations are poorly understood. Previously (Nee and D'Esposito, 2016; 2017), we detailed frontal dynamics that support control processing, but left open their role in broader cortical function. Here, I show that the FPCN consists of an external/present-oriented to internal/future-oriented cortical gradient extending outwardly from sensory-motor cortices. Areas at the ends of this gradient act in a segregative manner, exciting areas at the same level, but suppressing areas at different levels. By contrast, areas in the middle of the gradient excite areas at all levels, promoting integration of control processing. Individual differences in integrative dynamics predict higher level cognitive ability and amenability to neuromodulation. These data suggest that an intermediary zone within the FPCN underlies integrative processing that supports cognitive control.

**\*For correspondence:**
derek.evan.nee@gmail.com

**Competing interests:** The author declares that no competing interests exist.

## Introduction

While habits rigidly link stimuli to actions, cognitive control enables flexible behavior that can adapt to present conditions, prevailing contexts, and future plans (*Miller and Cohen, 2001*; *Logan and Gordon, 2001*; *Egner, 2017*; *Badre and Nee, 2018*). The human capacity for flexible behavior is thought to arise from the expansion of transmodal cortices that are synaptically distant from primary cortices (*Mesulam, 1998*; *Margulies et al., 2016*; *Huntenburg et al., 2018*) such that synaptic distance may untether transmodal areas from the processing hierarchies that link stimulus to action in canonical circuits (*Buckner and Krienen, 2013*). Such transmodal areas are prominent in the prefrontal (PFC) and posterior parietal cortices (PPC).

The relationship between cognitive control and areas of the PFC and PPC is so ubiquitous that a co-active set of PFC-PPC areas is frequently termed the 'frontoparietal control network' (FPCN) (*Vincent et al., 2008*; *Braga and Buckner, 2017*; *Yeo et al., 2011*; *Dixon et al., 2018*; *Murphy et al., 2020*). Recognizing their involvement across a diverse array of tasks, a similar constellation of areas is also referred to as the 'multiple demand network' (*Duncan and Owen, 2000*; *Duncan, 2010*; *Duncan, 2013*). This network is thought to implement cognitive control by flexibly coordinating activity among diverse brain systems to integrate brain-wide processing in a goal-directed manner (*Murphy et al., 2020*; *Cole et al., 2013*; *Cocchi et al., 2013*). The integrative capacity of these areas enables them to reconfigure the brain into difficult-to-reach states (*Gu et al., 2015*), thereby conferring the flexibility needed to act in an adaptive, rather than habitual manner. In this way, the integrative capacity of cognitive control is central to higher level cognition.

Mounting evidence suggests that there is not a single FPCN, but multiple networks (*Braga and Buckner, 2017*; *Yeo et al., 2011*; *Dixon et al., 2018*; *Murphy et al., 2020*). These networks are situated upon global brain gradients such that increasingly sensory-motor distal areas of the PFC and PPC are increasingly distant from areas of the brain involved in external processing (*Margulies et al., 2016*; *Huntenburg et al., 2018*). This has led to the proposal that the more sensory-motor distal aspects of the FPCN are involved in more internally oriented control processes

(*Dixon et al., 2018*; *Murphy et al., 2020*). The proposal that gradients in the PFC and PPC can be classed along an external-internal axis (*Sormaz et al., 2018*; *Buckner and DiNicola, 2019*) offers a unifying perspective of the different forms of cognitive control which has remained elusive (*Badre and Nee, 2018*). (A 'gradient' refers to a change in a property along an axis. With respect to cortex, a functional gradient means that functions vary in a roughly monotonic fashion with cortical distance along a spatial axis. The resolution of the gradient, or in other words, the steps along the axes, may be discrete and areal, or approach continuity. The present study remains silent regarding the resolution of these gradients, and takes the weak position that functions change along cortical axes at some unknown rate/resolution.)

Ultimately, cognitive control is grounded in behavior. This means that internally oriented aspects of control, such as those that plan for the future, must be integrated with externally oriented aspects of control, such as those that select appropriate sensory features for processing and action. How and where such integration takes place is an open question. One possibility is that the control gradient that expands outwardly from sensory-motor cortices doubles as an integration gradient such that the most sensory-motor distal areas are the most integrative. This possibility is consistent with theories of PFC function that posit that the rostral-most (i.e. most sensory-motor distal) areas act as apex controllers that exert widespread influences that can coordinate brain-wide activity under a single goal (*Badre and D'Esposito, 2009*). Another possibility is that a cascade of control signals progresses from sensory-motor distal to sensory-motor proximal areas with integration of those signals progressing along the way (*Koechlin et al., 2003*; *Koechlin and Summerfield, 2007*). A third possibility is that those areas situated between externally oriented and internally oriented control are responsible for their integration. This last possibility is consistent with recent data indicating the importance of mid-lateral PFC in integrative control (*Badre and Nee, 2018*; *Cocchi et al., 2014*; *Nee and D'Esposito, 2016*; *Nee and D'Esposito, 2017*). This latter possibility would suggest a nested structure to gradients in the brain: just as the FPCN is situated in intermediary zones of the brain to flexibly guide sensation to action, so too are intermediary zones of the FPCN essential for flexibly guiding control itself.

Here, two datasets that have demonstrated macroscale gradients of cognitive control in the PFC (*Nee and D'Esposito, 2016*; *Nee and D'Esposito, 2017*) are examined to investigate dynamics in the broader FPCN. These datasets employ a Comprehensive Control Task that manipulates three forms of cognitive control across two stimulus domains to precisely map the functional underpinnings of areas within the FPCN, providing a foundation from which to interpret network dynamics. The same gradients of activation observed in the PFC are mirrored in the PPC and tied to specific control functions and behavioral timescales. Interactions among areas of the PFC and PPC are measured by examining both static and dynamic indices of effective connectivity, revealing how control integrates processing to support adaptive behavior. Finally, static and dynamic measures of integration are used to predict trait-level cognitive ability, and susceptibility of cognitive control to neuromodulation showing the importance of integration for higher-level cognition and interventions more broadly. Collectively, these analyses elucidate the integrative organization of the FPCN whose insights may be useful to understand other transmodal networks.

## Results

Two independent samples (n = 24, n = 25) completed a Comprehensive Control Task that independently manipulated demands on stimulus domain (verbal vs spatial processing), sensory-motor control (associating stimuli to actions), contextual control (context-dependence of appropriate stimulus-response associations), and temporal control (planning for the future). These processes can be classed along an external-internal continuum such that sensory-motor control acts upon the sensory environment, contextual control informs those actions based upon an internalized rule, and temporal control informs the rule based upon an additional internalized representation (*Figure 1*). These processes can be classed by timescale such that sensory-motor control specifies an action for the present stimulus, contextual control specifies a prevailing task context, and temporal control sustains a prospective memory for the future. More generally, the processes are classed by abstraction with sensory-motor control being the most concrete and temporal control being the most abstract. A

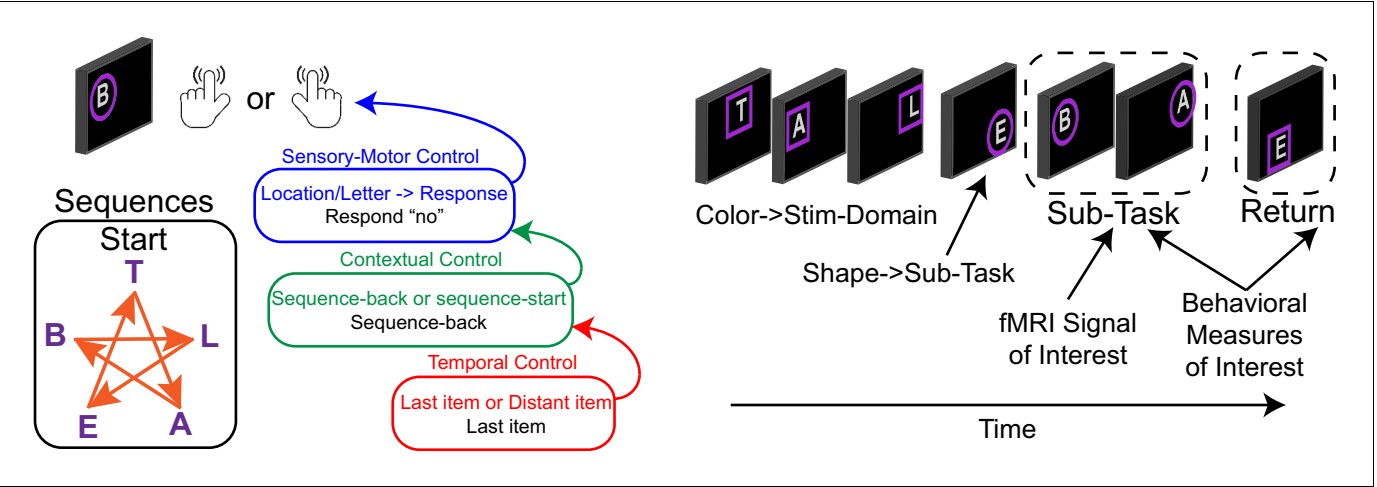

**Figure 1.** Comprehensive control task. On each trial, participants observed a letter at a spatial location and made a keypress in response. Keypresses mapped onto 'yes'/'no' responses. Participants either responded 'no' to the stimulus without regard to the stimulus features, or made a choice response based on a pre-learned sequence to a color cued feature (T-A-B-L-E-T for letters; star trace for locations) thereby engaging sensory-motor control. The correct response was either based upon a reference stimulus for which participants responded regarding whether the present stimulus followed a reference stimulus (sequence-back), or whether the stimulus was the start of the sequence (sequence-start). Switching among these tasks engages contextual control. Finally, the reference stimulus could either be the last item presented, or a distant item. In the latter case, the reference item had to be sustained over several trials requiring temporal control to prepare for the future. Colored frames indicated relevant stimulus features (letter or location), whereas frame shapes indicated cognitive control demands which were manipulated in the middle of each block (sub-task). Stimulus domain and cognitive control demands were independently manipulated in a factorial design wherein orthogonal contrasts separately isolated sensory-motor control, contextual control, and temporal control. Refer to the Materials and methods and *Nee and D'Esposito, 2016*; *Nee and D'Esposito, 2017* for a more complete description.

more complete description of the task can be found in the Materials and methods, and the original reports of these data (*Nee and D'Esposito, 2016*; *Nee and D'Esposito, 2017*). Behavioral data are reported in Supplemental Material.

The analyses that follow proceed in two parts. First, areas within the FPCN are functionally mapped and related to behavior by contrasting activation and performance across different task conditions. Areas within the PFC have been previously described (*Nee and D'Esposito, 2016*; *Nee and D'Esposito, 2017*), and are re-depicted here to note parallels among the PFC and PPC. A prior study (*Choi et al., 2018*) suggested a mirrored organization of the PFC and PPC with respect to cognitive control which the analyses here seek to replicate and extend by relating activations directly to behavioral performance. These analyses are aimed to provide a functional foundation for network-based analyses to follow. Second, interactions among areas in the FPCN are examined to address the focus of the current study of how integrative dynamics within the FPCN support cognitive control. These dynamics are then related to individual differences in trait-level cognitive ability and amenability to neuromodulation to establish their importance more broadly.

## Mirrored control gradients in the PFC and PPC

*Figure 2A* depicts the voxel-wise, whole-brain activations for each cognitive control contrast. In the lateral PFC, activations progressed in a caudal to rostral fashion as a function of abstraction of cognitive control from sensory-motor control, to contextual control, to temporal control (*Nee and D'Esposito, 2016*; *Nee and D'Esposito, 2017*). In the PPC, activations progressed in a rostral to caudo-lateral fashion as a function of abstraction of cognitive control (*Choi et al., 2018*). That is, in both the lateral PFC and PPC, increasingly abstract control was associated with activations increasingly distant from sensory-motor cortices of the pre/post-central gyri.

Different cognitive control demands produced overlapping activations eliciting gradients along the lateral surface (*Figure 2B*). Although the extent of overlap depends upon preprocessing (e.g. smoothing, volumetric vs surface processing), similar gradients of activation were observed with reduced volumetric smoothing (*Figure 2—figure supplement 1*), as well as with minimal, surface-based smoothing (*Figure 2—figure supplement 2*). The mirrored gradients observed in the lateral

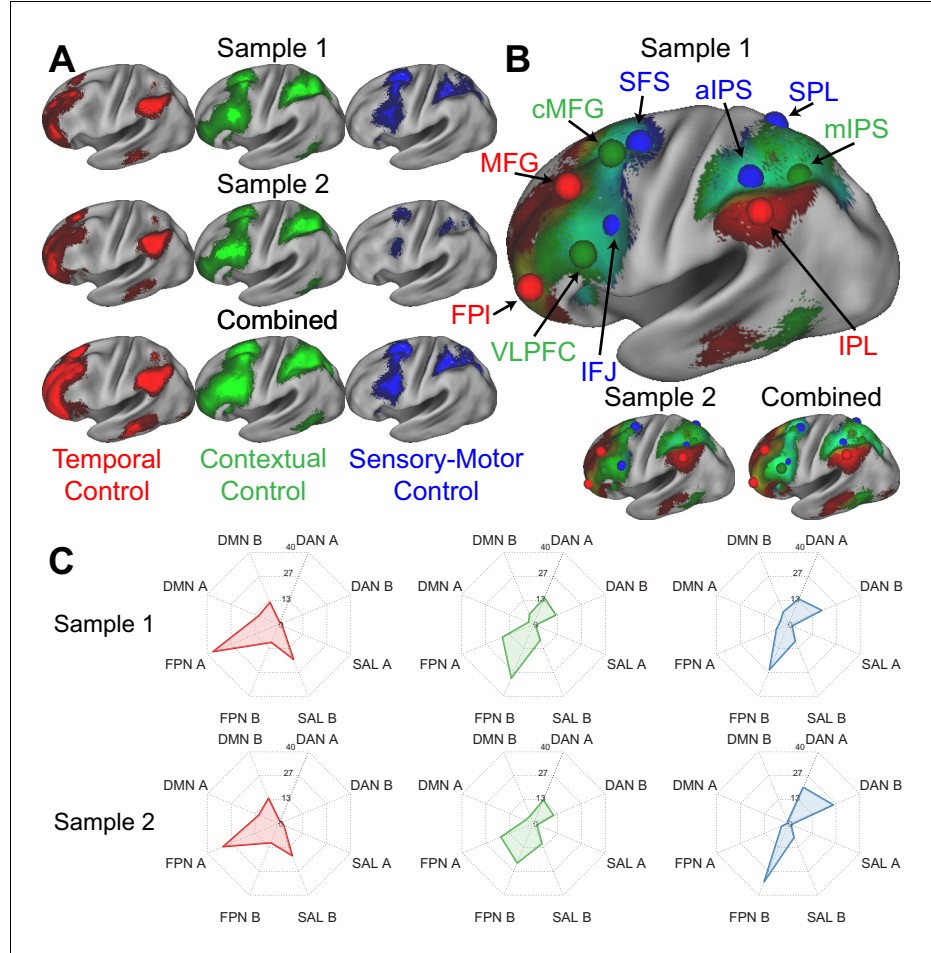

**Figure 2.** Control sub-networks. (**A**) Activations for temporal control (red), contextual control (green), and sensory-motor control (blue). Activations are depicted separately for each sample, as well as the samples combined. (**B**) Activations overlaid to demonstrate the gradient of cognitive control. Spheres indicate the location of regions-of-interest based upon activation peaks. (**C**) Overlap between activation contrasts (red – temporal control; green – contextual control; blue – sensory-motor control) and the 17-network parcellation described by Yeo et al. FPl – lateral frontal pole; MFG – middle frontal gyrus; VLPFC – ventrolateral prefrontal cortex; cMFG – caudal middle frontal gyrus; IFJ – inferior frontal junction; SFS – superior frontal sulcus; aIPS – anterior intra-parietal sulcus; IPL – inferior parietal lobule; mIPS – mid intra-parietal sulcus; SPL – superior parietal lobule; DMN – default-mode network; DAN – dorsal attention network; SAL – salience network; FPN – frontoparietal network.

The online version of this article includes the following figure supplement(s) for figure 2:

**Figure supplement 1.** Contrasts of temporal control (red), contextual control (green), and sensory-motor control (blue) and their overlap using reduced, 4 mm full-width half-maximum volumetric smoothing to reduce blurring.
**Figure supplement 2.** Contrasts of temporal control (red), contextual control (green), and sensory-motor control (blue) and their overlap using surface-based processing and reduced smoothing to minimize blurring.

PFC and PPC are suggestive of a mirrored functional organization extending outwardly from sensory-motor areas consistent with macroscale gradients of cortical function (*Mesulam, 1998*; *Margulies et al., 2016*; *Huntenburg et al., 2018*).

Previous work has suggested that the FPCN can be fractionated into sub-networks (*Braga and Buckner, 2017*; *Yeo et al., 2011*; *Dixon et al., 2018*; *Murphy et al., 2020*). In particular, it has been proposed that an FPCN network A acts as an intermediary between the FPCN and internally oriented 'default-mode' network (DMN). On the other hand, FPCN network B is hypothesized to act as an intermediary to the externally-oriented dorsal attention network (DAN) (*Dixon et al., 2018*; *Murphy et al., 2020*). However, precise functional descriptions of these sub-networks remains unclear. This is because prior work has fractionated the FPCN based on co-

activation patterns rather than processing demands. A marked benefit of the Comprehensive Control Task is the ability to dissociate multiple control processes in a single, well-controlled paradigm. To examine how the task contrasts align with co-activation defined networks, the overlap among activations and the *Yeo et al., 2011* 17-network parcellation was computed. In addition to considering sub-networks of the FPCN, other relevant sub-networks were also considered including the DMN, DAN, and salience networks (SAL). Across both samples, areas activated by temporal control aligned most closely with FPCN A (*Figure 2C*). By contrast, areas activated by both contextual control and sensory-motor control aligned most closely with FPCN B. While areas activated by contextual control also activated FPCN A to some degree, areas activated by sensory-motor control activated the DAN more extensively. Collectively, these data suggest that the task contrasts fractionate the FPCN into sub-networks along an external (DAN -> FPCN B) to internal (FPCN A -> DMN) axis. Furthermore, these observations offer more precise functional descriptions of previously fractionated sub-networks.

To better characterize the functions of the areas within these sub-networks, spherical regions-of-interest (ROIs) were centered on activation peaks of the contrasts (*Figure 2B*; *Table 1*). Activations across the eight conditions of the factorial design are depicted in *Figure 3A* providing a graphical profile of the activation patterns in each area. PFC-PPC areas closer to sensory-motor cortex (blue, green) tended to be sensitive to multiple demands activating for both sensory-motor and contextual control. These areas also had a tendency to activate preferentially to one stimulus domain or the other. By contrast, areas distal from sensory-motor cortices (red) tended to be more specialized toward temporal control and did not show a preference for stimulus domain.

To provide a more compact, data-driven description of these profiles, the PFC-PPC ROI data ((participants*conditions) x areas) were subjected to multi-dimensional scaling (MDS). Two dimensions accounted for 89% of the variance of the data (*Figure 3B*). The first dimension recapitulated the abstraction gradient, placing PFC-PPC areas proximal to sensory-motor cortices on one end of the dimension (e.g. superior frontal sulcus - SFS; anterior intra-parietal sulcus - aIPS), and areas distal to sensory-motor cortices on the other (e.g. middle frontal gyrus - MFG; inferior parietal lobule - IPL). Hence, across the different conditions of the task, this dimension collapsed abstraction of cognitive control into a single axis. The second dimension reflected sensitivity to stimulus domain with areas preferentially engaged by spatial processing at one end and areas preferentially engaged by verbal processing at the other. These patterns were also present with reduced smoothing (*Figure 3—figure supplement 1*). These data are consistent with the idea that the cortex is organized along two principle gradients reflecting abstraction and modality (*Mesulam, 1998*; *Margulies et al., 2016*;

**Table 1.** Coordinates of regions-of-interest reported in MNI space.

FPl – lateral frontal pole; MFG – middle frontal gyrus; VLPFC – ventrolateral prefrontal cortex; cMFG – caudal middle frontal gyrus; IFJ – inferior frontal junction; SFS – superior frontal sulcus; IPL – inferior parietal lobule; mIPS – mid intra-parietal sulcus; aIPS – anterior intra-parietal sulcus; SPL – superior parietal lobule.

| | Sample 1 | | | Sample 2 | | |
|---|---|---|---|---|---|---|
| Area | X | Y | Z | X | Y | Z |
| FPl | −44 | 48 | 4 | −36 | 52 | 0 |
| MFG | −38 | 28 | 44 | −36 | 34 | 38 |
| VLPFC | −52 | 20 | 28 | −42 | 30 | 20 |
| cMFG | −34 | 10 | 60 | −30 | 6 | 56 |
| IFJ | −38 | 6 | 26 | −42 | 10 | 24 |
| SFS | −22 | 0 | 54 | −20 | 0 | 56 |
| IPL | −54 | −50 | 44 | −56 | −52 | 42 |
| mIPS | −28 | −60 | 42 | −26 | −56 | 44 |
| aIPS | −34 | −40 | 46 | −30 | −42 | 42 |
| SPL | −14 | −52 | 64 | −12 | −60 | 58 |

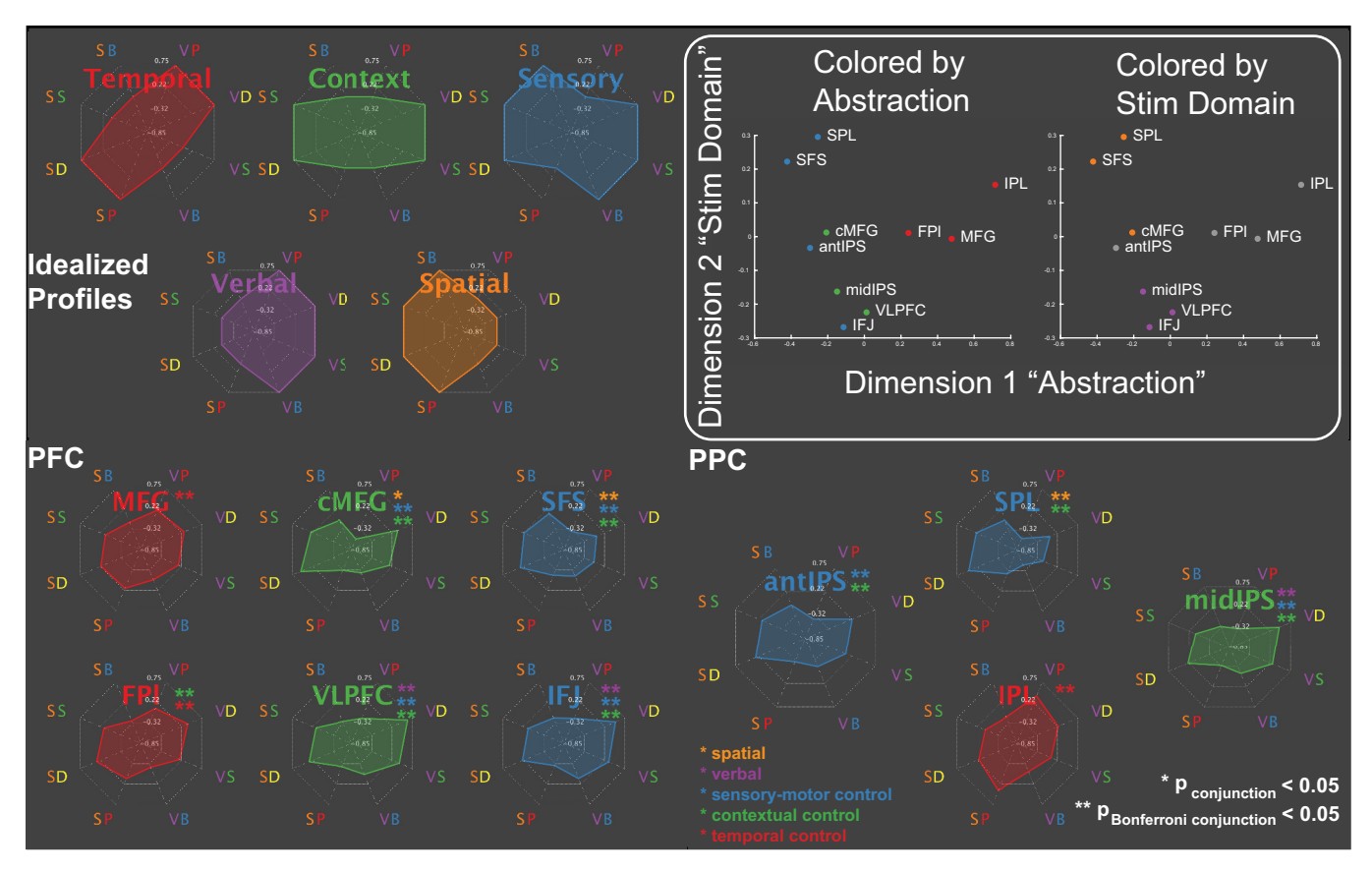

**Figure 3.** Activation profiles. Activations across the eight conditions of the task design are depicted as radar plots (SB – spatial baseline; VB – verbal baseline; SS – spatial switching; VS – verbal switching; SP – spatial planning; VP – verbal planning; SD – spatial dual; VD – verbal dual). The top panel depicts idealized profiles for areas sensitive solely to temporal control (red), contextual control (green), sensory-motor control (blue), verbal stimulus domain (purple), and spatial stimulus domain (orange). Inset: results of multi-dimensional scaling of the activation profiles across regions. Colored by abstraction refers to coloring as a function of position along the control gradient (blue – sensory-motor proximal; green – intermediary; red – sensory-motor distal). Colored by stim domain refers to coloring as a function of sensitivity to stimulus domains (orange – spatial; purple – verbal; gray – neither).

The online version of this article includes the following figure supplement(s) for figure 3:

**Figure supplement 1.** Activation profiles with reduced smoothing.

---

*Huntenburg et al., 2018*). Collectively, these dimensions provided a data-driven way to operationalize the factors of the task design.

## Control gradient is related to present-future behavior

Next, areas were characterized as a function of their relationship to behavior in the task. Activations were assessed during the sub-task phases that manipulated cognitive control demands (*Figure 1*). Cognitive control behaviors were expressed as trial-wise reaction times both during the sub-task phases, as well as during return trials that immediately followed the sub-task phase. For example, temporal control requires sustaining an internal representation (i.e. a reference stimulus) during the sub-task phase to be utilized on the return trial (i.e. does the stimulus on the return trial follow the reference stimulus in the sequence?). Therefore, activations during temporal control that sustain the internal representation would be expected to relate to behavior on the return trial (i.e. return trial reaction times), but not necessarily the sub-task trials (i.e. sub-task trial reaction times). Hence, activations can be separately correlated with behavior measured at the same time as the activations (sub-task trials: present behavior) or the behavior for which the activations are preparing (return trials: future behavior; see Materials and methods).

In the PFC, increasing rostral areas were increasingly associated with future behavior and decreasingly associated with present behavior (*Nee and D'Esposito, 2016*; *Nee and D'Esposito, 2017*). This pattern was mirrored in the PPC (*Figure 4A*) such that sensory-motor proximal areas (aIPS, SPL) were positively associated with present, but not future behavior, while sensory-motor distal areas (IPL) were positively associated with future, but not present behavior. Areas in-between (mIPS) were associated with both present and future behavior. These patterns were confirmed with voxel-wise analyses (*Figure 4B*). These data are consistent with an integrative role of contextual control areas positioned between internal, future-oriented temporal control areas and external, present-oriented sensory-motor control areas.

To better quantify these relationships, the abstraction dimension uncovered by MDS was used to account for areal relationships with behavior. Separate linear mixed effects models were fit for the activation-present behavior and activation-future behavior relationships across areas. Consistent with

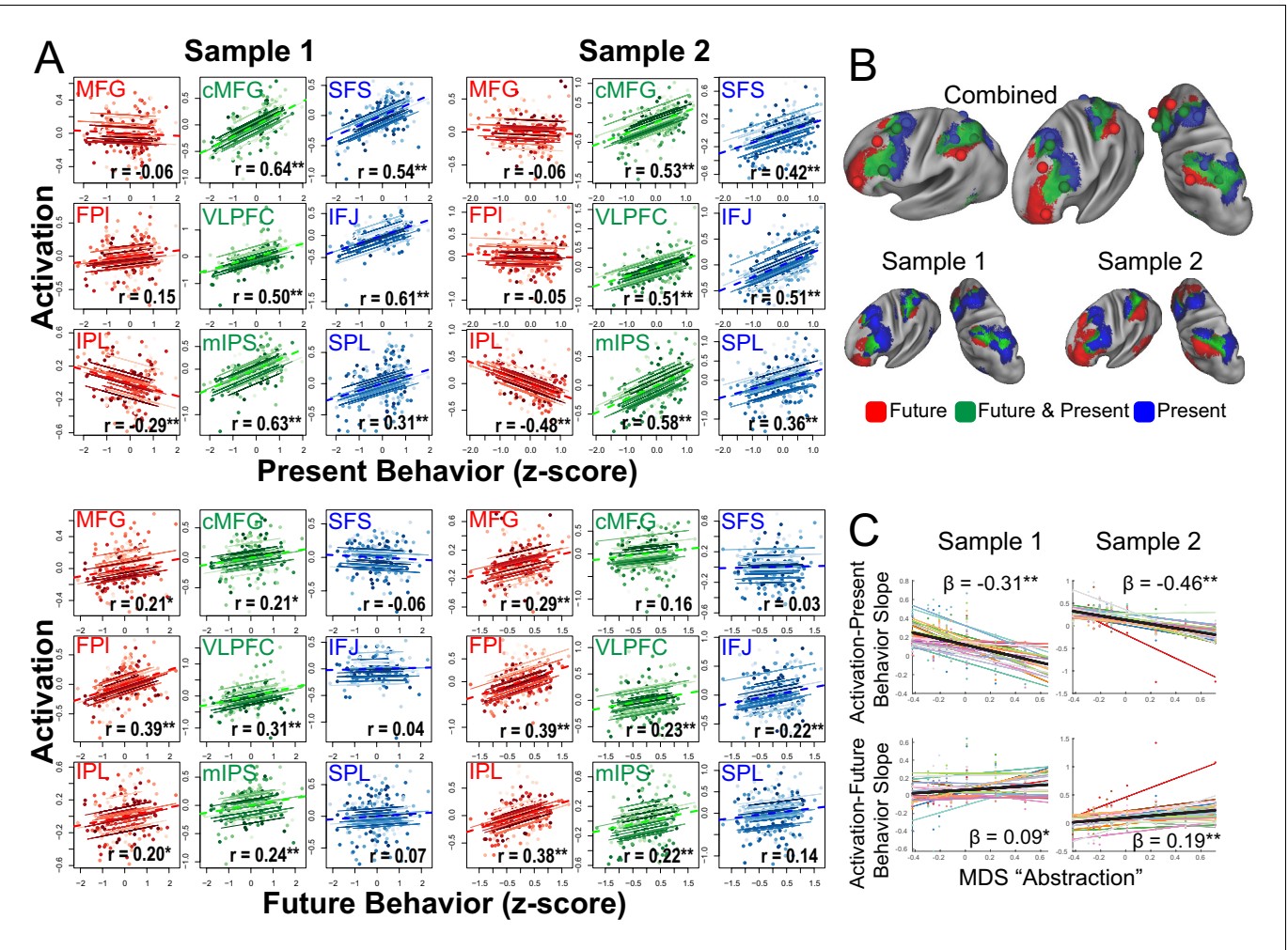

**Figure 4.** Brain-behavior relationships. (A) Top: Repeated measures correlations between activation and present behavior (i.e. behavior during sub-task trials; see *Figure 1*). Areas related to sensory-motor control (blue) and contextual control (green) showed positive associations, while areas related to temporal control (red) showed no or negative associations. Bottom: Repeated measures correlations between activation and future behavior (i.e. behavior during return trials; see *Figure 1*). Areas related to temporal control (red) and contextual control (green) tended to show positive associations, while areas related to sensory-motor control (blue) tended to show no associations. ** indicates Bonferroni-corrected p<0.05. * indicates uncorrected p<0.05. (B) Voxel-wise partial correlations between activation and present behavior (blue), future behavior (red), and both (green). Results are visualized at p<0.001 with 124 voxel cluster extent. (C) Linear mixed effects modeling of the activation-present behavior slope (top) and activation-future behavior slope (bottom) using the first dimension of multi-dimensional scaling (MDS) depicted in *Figure 3*. * indicates p<0.05; ** indicates p<0.005. The online version of this article includes the following figure supplement(s) for figure 4:

**Figure supplement 1.** Brain-behavior relationships with reduced smoothing.

the impressions above, the more abstract the PFC-PPC area, the less it related to current behavior (sample 1: t(238) = −9.18, p=2.19e-17; sample 2: t(248) = −8.73, p=3.81e-16) and more it related to future behavior (sample 1: t(238) = 2.72, p=0.007; sample 2: t(248) = 4.39, p=0.00002). These patterns were also present with reduced smoothing (*Figure 4—figure supplement 1*). Moreover, the abstraction-behavior relationships were observed when each stimulus domain was treated separately consistent with domain-generality of these patterns (verbal abstraction-current sample 1: t(238) = −6.61, p=2.52e-10; sample 2: t(248) = −8.11, p=2.38e-14; spatial abstraction-current sample 1: t (238) = −10.21, p=1.55e-20; sample 2: t(248) = −7.87, p=1.07e-13; verbal abstraction-future sample 1: t(238) = 3.05, p=0.003; sample 2: t(248) = 3.77, p=0.0002; spatial abstraction-future sample 1: t (238) = 2.29, p=0.01; sample 2: t(248) = 4.23, p=3.32e-5). Hence, these analyses indicate that progressively abstract areas in both the PFC and PPC are increasingly future-oriented.

## Establishing source-target relationships

Next, interactions among PFC-PPC areas were examined. Control is embodied by source-target relationships such that controllers affect processing in controlled targets. Effective connectivity, which estimates directed influences, offers the most straightforward means to assess such source-target relationships. There are both static (stationary) and dynamic (non-stationary) interactions among the PFC and PPC (*Cole et al., 2013*; *Cole et al., 2014*; *Krienen et al., 2014*; *Gratton et al., 2016*). To examine static interactions and their directed nature, a biophysically plausible generative model of how neuronal interactions produce cross spectra in the fMRI signal was employed (*Friston et al., 2014*; *Razi et al., 2015*; *Razi et al., 2017*).

The method was validated in two ways. First, the method was applied to the task data of both samples and estimates of directed interactions were compared across the samples. These estimates had excellent correspondence across the samples (r = 0.96; *Figure 5*, top). Second, estimates of directed interactions were used to examine the putative pre-eminent role of the PFC in cognitive control (*Miller and Cohen, 2001*). Cognitive control is exemplified by source-target relationships such that drivers of control asymmetrically influence targets. To characterize such asymmetries on

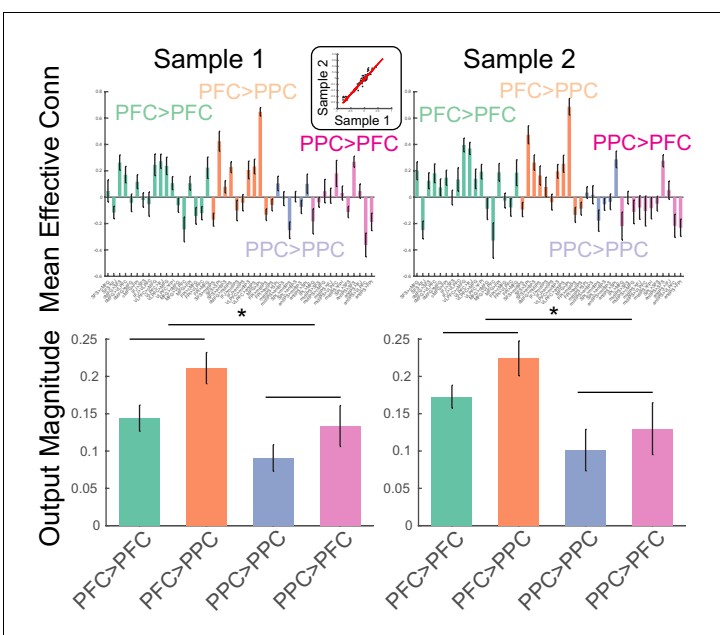

**Figure 5.** Estimates of static effective connectivity. Bars are color coded by source>target lobe pairs. Top: estimates for each connection. Inset: correlation among sample averaged parameter estimates. Bottom: estimates averaged over source>target lobe pairs. Influences arising from the PFC were significantly stronger in magnitude than those arising from the PPC. * indicates p<0.05.

The online version of this article includes the following figure supplement(s) for figure 5:

**Figure supplement 1.** Estimates of static effective connectivity with reduced smoothing.

the lobular level, the magnitude of effective connectivity (i.e. deviations from zero) were separately combined for each of the 2 × 2 combinations of source-target lobe (i.e. PFC->PFC, PFC->PPC, PPC->PFC, PPC->PPC). In both samples, the magnitude of effective connectivity arising from the PFC was significantly stronger than that arising from the PPC (sample 1: F(1,23) = 7.1, p=0.0138; sample 2: F(1,24) = 6.26, p=0.0195; *Figure 5*, bottom). Such data are consistent with the idea that the PFC is the primary driver of PFC-PPC dynamics. Collectively, these data indicate that the method produces replicable and theoretically sensible estimates.

## An intermediary integration zone of control

To examine integration of control signals, source-target relationships were assessed at the network level. Areas were assigned into networks based upon their position along the control gradient (i.e. as colored-coded in the analyses above; sensory-motor control network (blue): SFS, IFJ, SPL, aIPS; contextual control network (green): cMFG, VLPFC, mIPS; temporal control network (red): MFG, FPl, IPL). In both samples, a significant source x target network interaction was observed in effective connectivity (sample 1: F(4,46) = 25.1, p=4.56e-14; sample 2: F(4,48) = 17.81, p=5.74e-11; *Figure 6*). In all cases, within-network directed interactions were significant and numerically the most positive connections for each network. This is to be expected given that areas within a cortical network are assumed to excite one another, and also provides validation of the network assignment. For both nodes in the abstract, temporal control network and the concrete, sensory-motor control network, between-network directed interactions tended to be negative. Such patterns suggest that these networks dampen activity in other networks, thereby segregating processing. By contrast, between-network directed interactions arising from contextual control nodes were significantly positive (*Figure 6* – top panel). Control analyses revealed that these dynamics were not driven by task-related signals since regressing out task-related activity demonstrated the same pattern of results (*Figure 6—figure supplement 1*). Such patterns suggest that the contextual control network elevates activity in other networks, thereby promoting integrative processing.

*Figure 6* (middle panel) depicts these patterns in a more continuous form that does not depend upon assigning nodes to networks. Source-target relationships are depicted as a function of the abstraction dimension uncovered by MDS and described by the magnitude (marker size) and sign (marker shape: circle – positive, inverted triangle – negative) of effective connectivity. At the ends of the axis, positive effective connectivity is predominant when sources and targets are at a similar abstraction level (e.g. bottom left and top right), while negative effective connectivity is predominant when sources and targets are at different abstraction levels (e.g. top left and bottom right). By contrast, positive effective connectivity is observed throughout when sources are at the middle of the axis (i.e. columns of circles toward the middle of the horizontal axis). To better quantify these effects, linear mixed effects models were fit using the abstraction dimension uncovered by MDS. The model sought to explain effective connectivity as a function of abstraction of the source area, abstraction of the target area, and the interaction among source and target abstraction. Additionally, quadratic forms of each of these effects were included to capture U shaped and inverted-U shaped relationships. In both samples, a significant source x target abstraction interaction was observed (sample 1: t(1097) = 5.08, p=4.43e-07; sample 2: t(1143) = 4.94, p=9.03e-07) reflecting that network interactions depend upon whether sources and targets are drawn from similar/different levels of abstraction. Critically, there was also a quadratic effect of source abstraction (sample 1: t(1097) = −4.51, p=7.05e-06; sample 2: t(1143) = −4.52, p=6.99e-06), which was driven by areas at mid-levels of abstraction having consistently positive outwards influences, while areas at high and low levels of abstraction demonstrated both positive and negative outward influences which yielded lower net influences when averaged (*Figure 6* – bottom panel). No other effects were significant across samples. The same pattern of results was observed with reduced smoothing (*Figure 6—figure supplement 2*). Collectively, these patterns are consistent with the segregation/integration patterns described above and show that these patterns do not strictly depend on network assignment.

## Dynamic integration by contextual control

The static directed interactions described above indicate the *potential* for integration/segregation of the PFC-PPC during cognitive control. Actualization of this potential would require the activation of particular networks and corresponding activity flow (*Cole et al., 2016*). That is, activation of

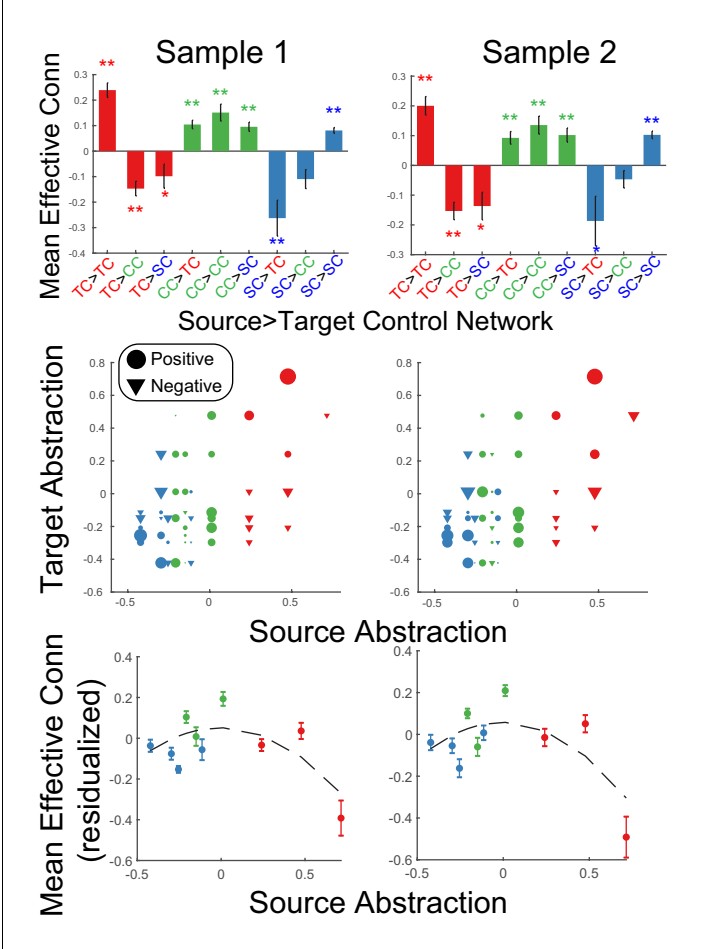

**Figure 6.** Static network interactions. Top: Effective connectivity was averaged as a function of source>target network. Bars are colored as a function of the source network: TC – temporal control (red); CC – contextual control (green); SC – sensory-motor control (blue). ** denotes Bonferroni-corrected p<0.05. * denotes p<0.05 uncorrected. Middle: Effective connectivity organized by abstraction (first dimension of multi-dimensional scaling depicted in *Figure 3*). Circles denote positive interactions and inverted triangles denote negative interactions. Markers are scaled by the magnitude of effective connectivity. Markers are colored by the network assignments. Bottom: Data re-depicted to highlight the quadratic effect of source abstraction. Linear effects of target abstraction, source abstraction, and target x source abstraction have been regressed out to isolate the quadratic effect of source abstraction demonstrating that mean (positive) effective connectivity peaks at areas in the middle of the abstraction gradient.

The online version of this article includes the following figure supplement(s) for figure 6:

**Figure supplement 1.** Effect of task signals on effective connectivity parameters and source x network interactions.

**Figure supplement 2.** Static network interactions with reduced smoothing.

---

intermediary, contextual control nodes should produce more integrated network interactions. To examine such actualization, psychophysiological interaction (PPI) analysis (*Cole et al., 2013*; *Friston et al., 1997*) was performed to estimate changes in effective connectivity as a function of different cognitive control demands.

Validation of the method by comparing estimates across samples (*Figure 7—figure supplement 1*) indicated that the context-independent effective connectivity estimates were replicable (r = 0.86), PPI's of temporal control showed modest replicability (r = 0.32), and PPI's of contextual control showed strong replicability (r = 0.78). PPI's of sensory-motor control were not replicable (r = −0.06). However, those areas engaged by sensory-motor control could largely be recapitulated by contrasts of stimulus domain (*Figure 7—figure supplement 2*). PPI's of stimulus domain showed strong

replicability (r = 0.88). Therefore, PPI's of stimulus domain were used as a proxy of interactions generated by activation of sensory-motor control areas.

For each cognitive control demand, source-target relationships among PFC-PPC areas were estimated. In this case, the interactions reflect the changes in effective connectivity induced by the cognitive control demands. In contrast to the stationary dynamics, within-network PPI's were generally weak across demands (*Figure 7A,B*). This indicates that within-network interactions are weakly modulated by task demands, which has been previously been observed in functional connectivity (*Cole et al., 2014*; *Krienen et al., 2014*; *Gratton et al., 2016*). Stronger modulations were observed between networks which was especially prominent during contextual control. To quantify these effects, within- and between-network interactions were separately averaged for each PPI contrast. These were then submitted to a 2 × 3 ANOVA with factors of network connectivity (within, between) and contrast (temporal, contextual, stimulus domain; *Figure 7B*). This analysis revealed a main effect of network connectivity (sample 1: F(1,23) = 15.01, p=0.0008; sample 2: F(1,24) = 10.34, p=0.0037), driven by stronger between- than within-network connectivity. There was no main effect of contrast (both samples p>0.15). However, there was a network connectivity x contrast interaction (sample 1: F(2,46) = 11.84, p=0.0001; sample 2: F(2,48) = 9.86, p=0.0003). This interaction was driven by stronger between-network connectivity for the contextual control contrast relative to the other contrasts. These results demonstrate that contextual control is associated with integration across PFC-PPC networks (*Figure 7C*), a pattern that was also observed with reduced smoothing (*Figure 7—figure supplement 3*).

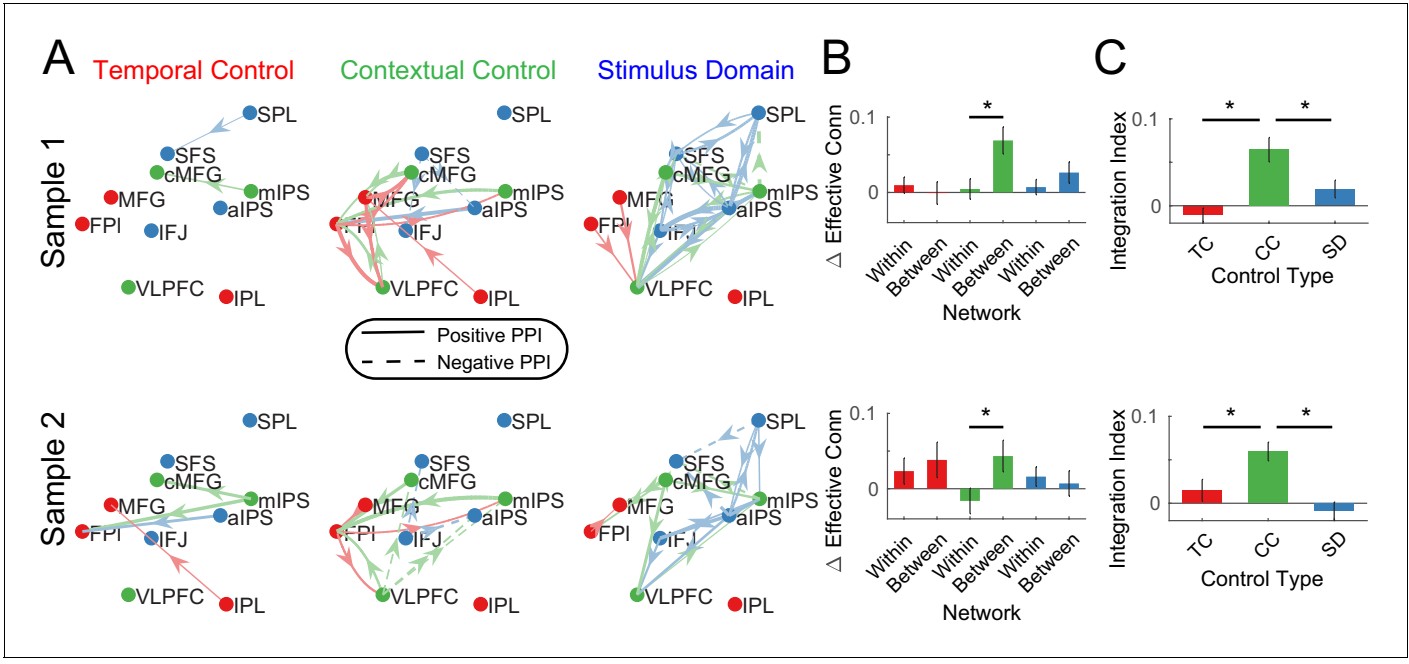

**Figure 7.** Dynamic Interactions. (**A**) Psychophysiological interactions (PPIs) for temporal control, contextual control, and stimulus domain contrasts. Solid lines indicate positive modulations of effective connectivity while dashed lines indicate negative modulations of effective connectivity. Arrows are colored by the network of the source node. Thickness of the arrows indicates the strength of modulations. Modulations are visualized at p<0.05 uncorrected. (**B**) Averaged within- and between-network modulations for the temporal control (red), contextual control (green), and stimulus domain (blue) contrasts. (**C**) Integration indices computed by contrasting between- minus within-network modulations for temporal control (TC), contextual control (CC), and stimulus domain (SD) contrasts.

The online version of this article includes the following figure supplement(s) for figure 7:

**Figure supplement 1.** Sample averaged psychophysiological interaction (PPI) parameter estimates.
**Figure supplement 2.** Comparison of the sensory-motor control contrast and stimulus domain contrasts.
**Figure supplement 3.** Dynamic interactions with reduced smoothing.

## Static and dynamic integration relates to higher level cognitive ability

The analyses above indicate that a PFC-PPC network involved in contextual control statically integrates processing among PFC-PPC networks, and that PFC-PPC networks are also more dynamically integrated when contextual control is required. Next, the relationship between individual differences in these network interactions (*Figure 8A,B*) and higher-level cognitive ability was examined. Each participant completed several tasks measuring short-term memory, working memory, and fluid intelligence. These measures were combined using principle components analysis (PCA) to form a composite measure of higher-level cognitive ability (*Figure 8C*). Measures of static and dynamic integration were then used to predict higher-level cognitive ability using two-fold cross-validated ridge regression. The static integration measure was intended to capture individual differences in

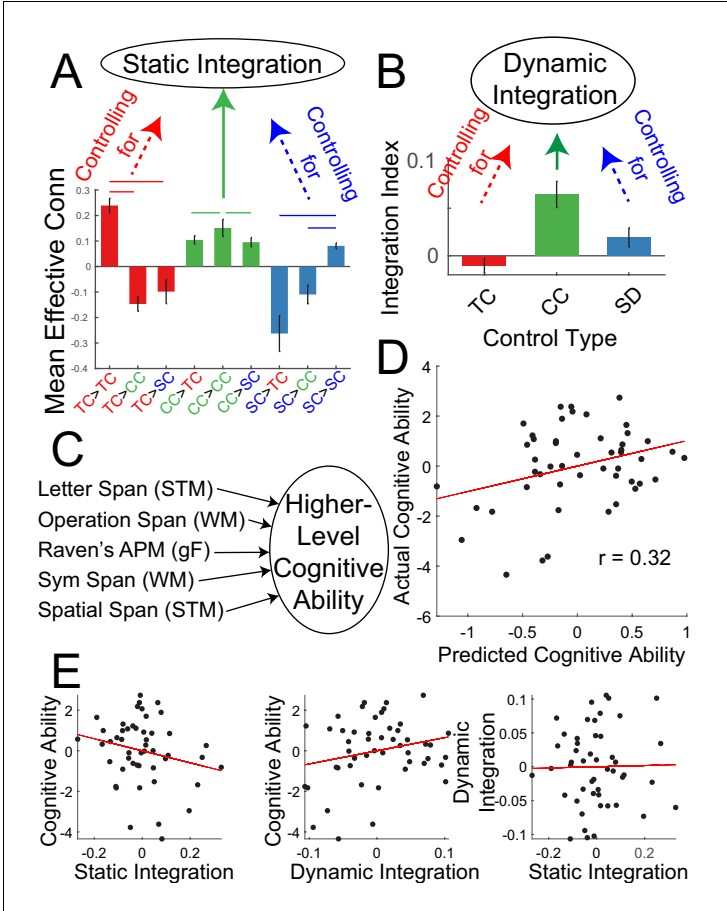

**Figure 8.** Integration predicts higher level cognitive ability. (A) Static integration was computed by contrasting between minus within-network effective connectivity of the contextual control network while controlling for (regressing out) the same contrast of the temporal control network and sensory-motor control network and individual differences in head motion. (B) Dynamic integration was based upon the integration index of the contextual control PPI's while controlling for (regressing out) the integration indices of the temporal control and stimulus domain PPI's and individual differences in head motion. (C) Higher level cognitive ability was computed as the first principle component of a battery of cognitive tests measuring capacity. (D) Cross-validated ridge regression was used to predict cognitive capacity based upon static and dynamic integration. The scatterplot depicts correlations between the predicted cognitive ability and actual cognitive ability. (E) Correlations among static integration, dynamic integration, and cognitive ability.

The online version of this article includes the following figure supplement(s) for figure 8:

**Figure supplement 1.** Activations predict higher level cognitive ability.

**Figure supplement 2.** Integration does not predict higher-level cognitive ability with reduced smoothing.

**Figure supplement 3.** With reduced smoothing, activation-based measures remain significantly predictive of higher-level cognitive ability.

the degree to which contextual control areas integrate PFC-PPC processing in a stationary manner. In other words, this measure captured the general integration of the FPCN without regard to specific demands. By contrast, the dynamic integration measure was intended to capture individual differences in the degree to which the FPCN became more integrated when contextual control was required. That is, this measure reflects integration 'on demand'. The data were separated by sample with one sample used to estimate regression weights, and the estimated regression weights used to predict higher-level cognitive ability in the other sample.

A significant prediction effect was observed (r = 0.32, p=0.029; *Figure 8D*) indicating that individual differences in integration dynamics are related to higher level cognitive ability. While static integration was associated with lower cognitive ability, dynamic integration was associated with higher cognitive ability (*Figure 8E*). These data indicate the importance of control network integration for higher level cognition.

It could be the case that network integration measures are superior to, redundant with, or complementary to activation-based measures for the purpose of predicting cognitive ability. To examine this question, factors were created to summarize activation-based metrics of temporal control (first principle component of the temporal control contrast in FPl, MFG, and IPL), contextual control (first principle component of the contextual control contrast in VLPFC, cMFG, and midIPS), sensory-motor control (first principle component of the sensory-motor control contrast in IFJ, SFS, antIPS, and SPL) and stimulus domain (first principle component of the stimulus domain contrast in IFJ, SFS, antIPS, and SPL). Similar to the integration measures, activation measures were able to predict individual differences in higher-level cognitive ability (r = 0.37, p=0.01; *Figure 8—figure supplement 1*). However, activation-based measures were complementary to network integration measures as evidenced by insignificant shared variance amongst the measures (activations explained ~3% of variance in the integration measures, adjusted R-square <0). Moreover, adding activation measures to the network integration measures significantly improved prediction accuracy (r = 0.50, p=0.002; nested model comparison using ordinary regression: F(4,42) = 2.84, p=0.048). Interestingly, whereas integration measures focused on static integration of the contextual control network, and dynamic integration during contextual control, activation during contextual control was the least informative of the activation-based measures (<1% explained variance; see *Figure 8—figure supplement 1*). Such data indicate that both network-based and activation-based measures can be useful for explaining cognitive ability, but do so in complementary manners.

The same analyses were repeated with reduced smoothing. Although dynamic integration was again positively associated with cognitive ability, with reduced smoothing, no relationship was observed among cognitive ability and static integration. As a result, integration measures could not be used to predict cognitive ability (r = 0.03; p=0.44; *Figure 8—figure supplement 2*). On the other hand, activation based measures remained significantly predictive (r = 0.31, p=0.04; *Figure 8—figure supplement 3*) and significantly improved prediction when added to models containing integration measures alone (r = 0.32, p=0.03; nested model comparison using ordinary regression: F(4,42) 3.05, p=0.04). I return to the implications of the non-replication with reduced smoothing in the Discussion.

## Static and dynamic integration relates to transcranial magnetic stimulation susceptibility

The fMRI data for sample two was used to localize targets for transcranial magnetic stimulation (TMS). TMS was performed on nodes in each sub-network: FPl – temporal control; VLPFC – contextual control; SFS – sensory-motor control, as well as a control site (S1). As previously reported (*Nee and D'Esposito, 2017*), we anticipated behavioral effects on temporal control following FPl stimulation, stimulus domain processing following SFS stimulation, and a contextual control x stimulus domain processing interaction following VLPFC stimulation. These expectations were observed (*Nee and D'Esposito, 2017*). However, individuals varied in their susceptibility to these effects. To examine whether such susceptibilities were related to the organization of cognitive control networks, the static and dynamic integration measures described above were used to predict TMS effects.

A PCA on the TMS effects was performed to derive a general susceptibility of cognitive control to PFC TMS (*Figure 9A*). Static and dynamic integration measures were used to predict the cognitive control TMS effect using leave-1-out cross-validated ridge regression. This analysis revealed that TMS effects on cognitive control could be predicted by static and dynamic integration (r = 0.56,

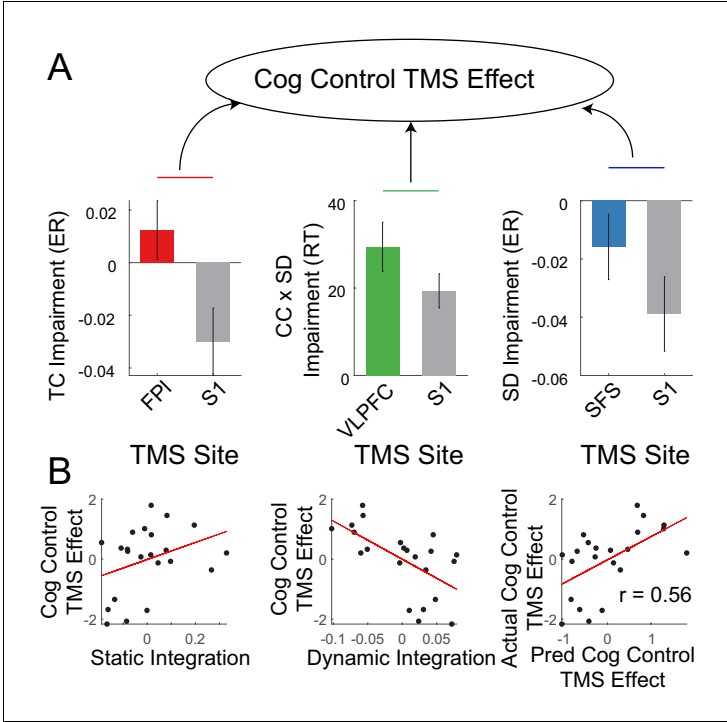

**Figure 9.** Integration predicts transcranial magnetic stimulation (TMS) effects. (**A**) Previously reported *Nee and D'Esposito, 2017* cognitive control impairments induced by continuous theta-burst TMS were combined using principle components analysis to derive a general susceptibility to PFC TMS. (**B**) Correlations between cognitive control TMS effects and static and dynamic integration. Leave-one-out cross-validated ridge regression was used to predict TMS effects using static and dynamic integration. Scatterplot depicts the correlation between predicted and actual TMS impairments.

The online version of this article includes the following figure supplement(s) for figure 9:

**Figure supplement 1.** Integration predicts transcranial magnetic stimulation (TMS) effects with reduced smoothing.

**Figure supplement 2.** Integration predicts transcranial magnetic stimulation (TMS) effects after controlling for cognitive ability.

**Figure supplement 3.** Activations do not predict TMS effects.

**Figure supplement 4.** Integration predicts transcranial magnetic stimulation (TMS) effects after controlling for both cognitive ability and activations.

p=0.01; *Figure 9B*). In particular, those individuals with stronger static integration tended to show increased TMS-induced cognitive control impairments, while those individuals with stronger dynamic integration tended to show decreased TMS-induced cognitive control impairments. These results were replicated with reduced smoothing (*Figure 9—figure supplement 1*). Such data indicate that integration of control networks is related to susceptibility to neuromodulation. Follow-up analyses suggested that only domain-general, but not domain-specific susceptibilities to TMS could be predicted (see Supplemental results).

Next, additional relationships among TMS susceptibility, cognitive ability, and activation were examined. TMS susceptibility and cognitive ability showed a non-significant negative relationship (r = −0.33, p=0.13). To examine whether integration measures predicted TMS susceptibility over-and-above cognitive ability, cognitive ability was regressed out of measures of static and dynamic integration. The resultant measures could be used to predict TMS susceptibility (r = 0.46, p=0.02; *Figure 9—figure supplement 2*). These data suggest independence between prediction of cognitive ability and TMS susceptibility. Next, activation-based measures were used to predict TMS susceptibility similar to the procedures detailed above. Unlike cognitive ability, activation-based measures could not predict TMS susceptibility (correlations among predicted and observed TMS effects all <0; *Figure 9—figure supplement 3*). Moreover, after regressing out both cognitive ability

and activation measures from static and dynamic integration, integration measures continued to predict TMS susceptibility (r = 0.47, p=0.02; *Figure 9—figure supplement 4*). These effects were all replicated with reduced smoothing (*Figure 9—figure supplements 2–4*). Hence, these data suggest that network integration is useful for predicting susceptibility to neuromodulation over-and-above cognitive ability and control-related activations.

## Discussion

The FPCN is thought to support cognitive control through integrating diverse networks (*Murphy et al., 2020*; *Cole et al., 2013*; *Cocchi et al., 2013*). However, evidence that the FPCN itself is not unitary, but fractionates into multiple sub-networks (*Braga and Buckner, 2017*; *Yeo et al., 2011*; *Dixon et al., 2018*; *Murphy et al., 2020*) leaves open questions of the functional roles of these sub-networks, and how their interactions support cognitive control. Here, functional mapping of the FPCN was established by contrasting multiple forms of control within a single paradigm. This revealed a mirrored organization in the PFC and PPC. More concrete forms of control that placed demands on stimulus-action selection (sensory-motor control) activated areas proximal to sensory-motor cortices. Increasingly more abstract forms of control which guided stimulus-action selection based upon prevailing contexts (contextual control), and temporally extended internal representations (temporal control) activated areas increasingly distal from sensory-motor cortices. Through correlating activation magnitudes with behavioral reaction times, it was observed that activations in areas responsive to sensory-motor and contextual control rose with reaction times in the moment, consistent with the idea that these areas are engaged commensurate with externally driven demands. By contrast, activation in areas sensitive to contextual and temporal control rose with future behavioral reaction times, consistent with a role of these areas in sustaining internal representations to prepare for future demands. Collectively, these data suggest that the FPCN is organized along a present/external to future/internal axis extending from sensory-motor proximal to sensory-motor distal areas.

Analyses on effective connectivity (directed interactions) examined network dynamics supporting cognitive control. These analyses revealed the existence of an integrative set of control areas whose dynamics relate to higher level cognitive ability and amenability to neuromodulation. These integrative areas were situated in an intermediary zone of the mirrored PFC-PPC gradient. Networks on either end of this control gradient acted in a segregative manner, exciting within-network nodes, while suppressing between-network nodes. Such dynamics may support selective processing of one time horizon (present/future) or medium (external/internal) at the exclusion of another. By contrast, the integration of both external/present-oriented and internal/future-oriented control networks may support hierarchical control wherein appropriate behaviors are jointly contingent on external and internal representations. The broader importance of such integrative processing was underscored by the relationship of individual differences in integrative dynamics and higher-level cognitive ability on the one hand, and amenability to neuromodulation on the other. Hence, the dynamics may be useful to specify optimal cognitive function, and predict responsivity to interventions.

### Beyond the PFC: toward a network view of cognitive control

A substantial body of work has focused on the functional organization of the lateral PFC and interactions therein that support cognitive control. Despite long-standing recognition that cognitive control is supported by areas distributed across the frontal and parietal lobes (*Vincent et al., 2008*; *Yeo et al., 2011*; *Duncan, 2010*; *Cole et al., 2013*; *Cole and Schneider, 2007*; *Power and Petersen, 2013*; *Fedorenko et al., 2013*; *Warren et al., 2014*; *Duncan et al., 2020*), much work, particularly in the domain of hierarchical cognitive control, has centered narrowly on how processing varies along the rostral-caudal axis of the PFC (*Badre and Nee, 2018*; *Badre and D'Esposito, 2009*; *Koechlin et al., 2003*; *Nee and D'Esposito, 2016*; *Nee and D'Esposito, 2017*; *Badre and D'Esposito, 2007*; *Badre, 2008*). Although a number of insights have been gained by focusing on the PFC, such a narrow focus ignores the broader networks in which the PFC participates. *Choi et al., 2018* recently demonstrated that the same rostral-caudal organization for control observed in the PFC is reflected in the PPC consistent with the idea that the PFC and PPC are comprised of ordered networks for control. The data here replicate those findings while also linking PFC-PPC activations with behavior along distinct timescales. Hence, it appears to the case that many functions that have been

attributed to the PFC are also present in the PPC. This necessitates expanding the study of cognitive control beyond the PFC to network and brain-wide levels.

## Macroscale gradients and cognitive control

There has been a recent surge of interest in macroscale gradients across the cortex (*Margulies et al., 2016*; *Huntenburg et al., 2018*; *Wang, 2020*). This interest stems from the thought that gradients provide a scaffold for functional processing thereby offering a window into how large-scale networks support cognition. Following *Mesulam, 1998*, *Margulies et al., 2016* proposed the existence of two macroscale axes upon which gradients are built with the first reflecting connectivity distance from primary cortices and the second reflecting modality. *Huntenburg et al., 2018* further proposed that the first gradient matches a temporal gradient reflecting the fundamental timescale over which a cortical area operates (*Hasson et al., 2015*; *Chaudhuri et al., 2015*; *Murray et al., 2014*), and the abstractness of the mental representations processed by the cortical area. These ideas are similar to those proposed by *Fuster, 2001* who hypothesized that mirrored frontal and posterior abstraction gradients support the temporal organization of controlled behaviors. However, within these frameworks, it is unclear exactly whether or how control should be distinguished from processing. That is, if the cortex can be organized along a principle axis of abstraction, does control emerge at the far end of that axis, all throughout, or in-between?

The position of the FPCN with respect to macroscale gradients of cortex is intermediary between canonical circuits involved in modality-specific processing, and the default-mode network involved in internal mentation (*Buckner and DiNicola, 2019*; *Raichle et al., 2001*). The intermediary positioning of the FPCN may be optimal for the orchestration of diverse brain regions in the service of adaptive behavior (*Cole et al., 2013*). Consistent with prior data (*Braga and Buckner, 2017*; *Yeo et al., 2011*; *Dixon et al., 2018*; *Murphy et al., 2020*), the data here indicate that the FPCN is not unitary but itself has a gradient organization. In particular, the data suggest that an intermediary zone of the FPCN, which itself is an intermediary zone of macroscale cortical gradients, is involved in the integration of control networks for adaptive behavior. Such data suggest the importance of examining gradients at multiple scales to understand brain-wide function. For example, the default-mode network has also been revealed to be comprised of multiple networks (*Braga and Buckner, 2017*; *Yeo et al., 2011*; *Buckner and DiNicola, 2019*). Extrapolating from the findings here, default-mode network B may act as an intermediary between the memory-related default-mode network A and the FPCN, positioned to integrate memory with control to guide internally-oriented cognition. Consistent with this idea, although the areas engaged by temporal control most closely aligned with FPCN A, there was some overlap with default-mode network B, suggesting some shared function among FPCN A and default-mode network B. Further investigation of the interactions among these networks would be an interesting future endeavor.

One question that remains open is the putative functions of FPCN sub-networks. Prior data that have fractionated the FPCN into sub-networks have done so based on co-activation patterns (*Braga and Buckner, 2017*; *Yeo et al., 2011*; *Dixon et al., 2018*; *Murphy et al., 2020*) which leave open the precise functional role of FPCN sub-networks. On the one hand, the overlap of areas sensitive to temporal control and FPCN A is suggestive that FPCN A acts upon internal representations to prepare for future controlled processing. These internal representations are likely to be relayed from areas in the DMN (*Dixon et al., 2018*; *Murphy et al., 2020*) potentially providing abstract schematic representations that are useful for organizing control processes toward a goal (*Badre and Nee, 2018*). On the other hand, areas sensitive to sensory-motor control overlapped most strongly with FPCN B. FCPN B is thought to coordinate with the DAN (*Dixon et al., 2018*; *Murphy et al., 2020*). This relationship with the DAN is consistent with the idea that attentional selection is necessary to guide action toward task-relevant stimuli at the exclusion of task-irrelevant stimuli – processes that support sensory-motor control.

However, alignment between task activation contrasts and previous co-activation based parcellations was mixed such that each putative control process examined here activated multiple sub-networks. To some extent, this is to be expected given that task contrasts are rarely process pure. On the other hand, areas can shift network allegiances as a function of task demands (*Salehi et al., 2020a*), which is further compounded by the finding that what constitutes an area/parcel itself shifts as a function of task demands (*Salehi et al., 2020b*). Indeed, the variable connectivity of the FPCN is thought to be the essence of its control capacities (*Cole et al., 2013*). Moreover, control, by

definition, denotes a source-target relationship. Hence, it may make little sense to consider any sub-network in isolation as representative of a control process. Instead, constellations of network activations and connectivities may be the unit of function when considering cognitive control. Commensurate with this idea, temporal control was profiled by activations most strongly in FPCN A, but also in DMN B, indicative of control acting on internal representations. Sensory-motor control was profiled by activations most strongly in FPCN B, but also in DAN, indicative of control acting on external representations. Finally, contextual control was profiled by activations most strongly in FPCN B, but also in FPCN A indicative of a co-operation of internal and external control processes. Collectively, these data serve to functionally ground the sub-networks that have been identified through co-activations.

One question that remains open is the spatial resolution of macroscale cortical gradients. The gradients depicted here appear somewhat continuous (*Figure 2*). However, the appearance of such continuity could be the result of any number of processing steps including spatial interpolations, smoothing, and averaging. For example, discrete individual variability can appear smooth and continuous after group averaging (*Braga and Buckner, 2017*; *Gordon et al., 2017*). When examined in an individual, connectivity profiles have been demonstrated to abruptly shift along spatial axes (*Wig et al., 2014*) suggesting that the cortex is best considered to be composed of discrete areas. Moreover, a recent study examining the scale of gradients in the PFC of non-human primates also concluded that such gradients vary in an areal rather than smooth fashion (*Tan et al., 2020*). Such data are consistent with the longstanding idea that discrete cortical areas perform discrete functions (*Brodmann, 1909*; *Amunts and Zilles, 2015*). Because of the processing and spatial resolution of the data here, the present dataset is unable to speak to finer gradations that may exist below the areal level, and is not inconsistent with the idea that the resolution of macroscale gradients is the areal level.

## Integration and cognitive control

Previous work has indicated the presence of an integrative core of regions important for multiple tasks (*Duncan and Owen, 2000*; *Duncan, 2010*; *Duncan, 2013*; *Cole et al., 2013*; *Cocchi et al., 2014*; *Cole et al., 2012*; *Shine et al., 2019a*; *Dosenbach et al., 2008*; *Dubois et al., 2018*). For example, *Shine et al., 2019a* recently performed a spatiotemporal principle components analysis across multiple tasks of the Human Connectome Project. The first principle component of this analysis rose and fell in cadence with task demands across multiple domains with greater rising associated with greater fluid intelligence. Similarly, *Cole et al., 2012* found that the global brain connectivity of a region in mid-lateral PFC predicted individual differences in fluid intelligence. *Cocchi et al., 2014* found that a similar region showed enhanced dynamic coupling with a diverse set of brain areas as the complexity of reasoning demands increased. Collectively, these data indicate the need for integration across tasks and demands for higher level cognition. Our data situate the areas associated with such integration in an intermediary zone of the FPCN.

In contrast to the integrative dynamics of intermediary areas, sensory-motor proximal and distal areas acted in a segregative fashion. Segregation is likely to be important to select relevant information while suppressing irrelevant information. Consistent with the data here, *Shine, 2019b* posited that rostral areas of the PFC are involved in segregation while mid-lateral areas are involved in integration. He theorized that segregation is mediated by cholinergic modulations while integration is mediated by noradrenergic modulations. Further research into the influences of distinct PFC-PPC networks on neuromodulatory mechanisms would be fruitful to elucidate such effects.

The data demonstrated both static and dynamic forms of integration. Areas involved in contextual control tended to excite other control networks, providing a potential substrate for integration through binding. Moreover, demands on contextual control increased inter-network communication producing a dynamic form of integration. Both static and dynamic integration were related to individual differences in higher level cognitive ability. Interestingly, the different forms of integration tended to be associated with opposite effects. On the one hand, increased static integration was associated with decreased higher level cognitive ability. It has been posited that an appropriate balance of segregation and integration into distinct networks or modules is important to optimize brain efficiency (*Sporns, 2010*). In particular, it has been observed that more segregation between networks, and integration within networks (i.e. modularity) at rest is associated with memory capacity (*Stevens et al., 2012*). Therefore, integration across networks in a stationary manner may be suboptimal for cognition. On the other hand, increased dynamic integration was associated with

increased higher-level cognitive ability. These data are consistent with previous studies that have shown that integrative reconfigurations of brain networks from rest to task are related to improved performance on complex tasks (*Cohen et al., 2014*; *Braun et al., 2015*; *Cohen and D'Esposito, 2016*). These data suggest that the brain's ability to integrate 'on-demand' is beneficial to cognitive processing.

Areas and networks that bind together processing of multiple brain systems are integral to multi-modal processing. A potential cost of such an organization is that integral processing hubs convey a vulnerability to cognitive processing such that damage to key areas/networks will have widespread impacts (*Warren et al., 2014*; *Gratton et al., 2012*). The data here suggest that in the domain of control processing, static integrative dynamics provide increased susceptibility to TMS-induced control deficits. The form of TMS explored here was aimed to be inhibitory (*Huang et al., 2005*) to cause deficits in control processing. Many therapeutic approaches to TMS utilize excitatory protocols in order to enhance irregular hypo-function (e.g. *Blumberger et al., 2018*). It would be interesting to see whether the widespread impacts of stimulation to integrative nodes can enhance therapeutic effects when excitatory protocols are employed.

Although I have suggested an integrative role of areas positioned in the middle of the FPCN, other accounts are plausible. For example, the excitatory influences of the mid-positioned areas in the FPCN on areas at either end of the FPCN could be considered a form of top-down modulation which may serve to strengthen or sharpen representations (*Miller and Cohen, 2001*; *Miller and D'Esposito, 2005*). This perspective could entail a more focal or localist influence of the mid-positioned areas of the FPCN than the integrative role that has been suggested. While tenable, I favor an integrative view given the global impact on network architecture observed when hub-like areas are damaged (*Gratton et al., 2012*). Moreover, top-down control by the FPCN has typically been document by long-range connections upon those areas responsible for stimulus representation (*Feredoes et al., 2011*; *Miller et al., 2011*). However, this does not preclude a top-down biasing role of the FPCN upon itself. Resolving this matter would likely require additional data that can identify the distinct representations of different areas of the FPCN and how those representations are modified by network dynamics.

The FPCN is not the only network important for integration and cognitive control. Extensive work has indicated the importance of the cingulo-opercular network (*Cocchi et al., 2013*; *Dosenbach et al., 2008*; *Cohen et al., 2014*; *Marek et al., 2015*) . Moreover, a cognitive control organization that parallels the PFC has recently been observed in the cerebellum (*D'Mello et al., 2020*). In the cerebellum, as in here, intermediary areas involved in contextual control showed an integrative role, mediating relationships among control areas at either end of the external/internal, present/future axis. Detailing interactions among the FPCN and these other networks would provide valuable insights into the brain-wise basis of cognitive control.

## Limitations

Several limitations should be considered to properly contextualize the findings reported here. First, positive effective connectivity has been posited as an excitatory influence while negative effective connectivity has been posited as an inhibitory influence. However, such interpretations should be taken with caution. Although spDCM models the translation of synaptic activity to the blood-oxygenation level dependent (BOLD) signal, it does not attempt to model more specific cellular and molecular processes. Moreover, estimates of effective connectivity depend upon the areas modeled. That is, an estimated direct influence among areas A and B could potentially be mediated by an un-modeled area C. In particular, it could well be the case that negative influences among areas in the FPCN are mediated by interactions with other structures such as the thalamus and basal ganglia (*Frank and Badre, 2012*; *Frank et al., 2001*; *Badre and Frank, 2012*). Hence, these observations should be taken as a starting point from which more complex and biologically plausible networks can be expanded.

Second, conclusions regarding the relationships among individual differences in FPCN network integration and cognitive ability/susceptibility to neuromodulation should be tempered by the sample sizes studied. Estimated effect sizes can be inflated at small sample sizes giving the impression that more variance is explained by a given measure than is actually the case (*Yarkoni and Braver, 2010*). Moreover, estimates of brain-phenotype relationships can be unstable at small sample sizes leading to poor replicability (*Marek, 2020*). The use of multi-variate, cross-validated methods here

should improve replicability and generalizability (*Kragel et al., 2020*). However, replication with larger samples is needed to draw firm conclusions regarding the relationships among FPCN network integration, cognitive ability, and susceptibility to neuromodulation.

Third, multiple steps have been taken to ensure the replicability of the findings here including replicating results across independent samples, and repeating the analyses with different preprocessing choices. All but one reported finding replicated across analyses: the use of FPCN network integration measures to predict cognitive ability did not replicate with reduced smoothing. In light of the many replicated results, this non-replicated result warrants consideration. The lack of replication could indicate that the original finding was a false positive and/or that power is insufficient in the present sample to produce a stable effect size. Another possibility is that reduced smoothing limited the ability of brain-derived FPCN integration measures to predict behavior. Behavior arises from population-level dynamics such that considering more of the relevant population-level signals should facilitate better explanations of behavior. Hence, although less smoothing makes the signals more focal, which is good for dissociating nearby areas, it may be that pooling across nearby areas makes the signals more representative of the population-level dynamics that explain behavior. Toward this possibility, in all the analyses that used neural data to predict behavior, prediction accuracy, even when significant, was universally diminished with reduced smoothing. Whether the lack of replication of FPCN integration measures predicting cognitive ability with reduced smoothing indicates that the original result/analysis was a false positive/had insufficient power or whether this is a reflection of a true result varying as a function of preprocessing choices requires future investigation. For the time-being, this result should be taken with some measure of caution.

Fourth, the analyses presented here have focused exclusively on the left hemisphere. This is in keeping with prior work with these data (*Nee and D'Esposito, 2016*; *Nee and D'Esposito, 2017*) and is consistent with the preponderance of work in the domain of abstraction and hierarchical control showing more marked recruitment of the left hemisphere (*Badre and Nee, 2018*; *Badre and D'Esposito, 2007*; *Nee and Brown, 2012*; *Nee and Brown, 2013*; *Nee et al., 2014*; *Bahlmann et al., 2014*; *Bahlmann et al., 2015*; *Jeon and Friederici, 2013*; *Jeon et al., 2014*). The reason for this left lateralization is unclear, but may relate to the presumed development of internalized control structures via interactions with the language system (*Luria, 1959*; *Baddeley, 2012*). However, some control processes, such as those responsible for the inhibition of motor responses tend to be more right lateralized (*Aron et al., 2004*; *Aron, 2007*). Future work would do well to compare and contrast network integration of the left and right FPCN.

## Materials and methods

### Participants

Twenty-four participants (13 female; age 18–28, mean 19.9 years) formed sample 1, previously described (*Nee and D'Esposito, 2016*). Twenty-five participants (16 female; age 18–27; mean 20.6 years) formed sample 2, previously described (*Nee and D'Esposito, 2017*). Collectively, the total sample size was forty-nine.

All participants were screened to be right-handed, native English speakers or fluent by the age of 6, and had no reported history of neurological or psychiatric disorders. Informed consent was obtained in accordance with the Committee for Protection of Human Subjects at the University of California, Berkeley.

The targeted number of participants was based upon previous work with related paradigms. The empirical replicability of univariate activations from these data has been previously established (*Nee, 2019*) indicating that power is sufficient for the estimation of cognitive control networks at the group level at the collected sample sizes. To ensure replicability of more complex analyses, bi-variate, and effective connectivity analyses in the present study are reported separately for each sample when possible, providing indications of replicability and power sufficiency.

### Experimental design and statistical analyses
#### Comprehensive control task
The Comprehensive Control Task was adapted from prior work (*Koechlin et al., 1999*; *Charron and Koechlin, 2010*), and designed to manipulate multiple forms of cognitive control within a single,

well-controlled paradigm. These different forms of control can be classed by different levels of abstraction with concrete processing acting on external stimuli in the present moment, and abstract processing supporting the maintenance of internal representations to guide future behavior. At the lowest level, *sensory-motor control* selects relevant stimulus features and associated actions. At the mid-level, *contextual control* selects the rules that guide the appropriate stimulus-response associations. Finally, at the highest level, *temporal control* supports temporally-extended representations that prepare for future control demands.

On each trial of the task (*Figure 1*), participants were presented with a letter at one of five spatial locations surrounded by a colored shape. Each stimulus required a choice decision indicated by a left or right keypress. Choice-to-keypress mappings were counter-balanced between participants. The correct decision depended upon a combination of (1) a stimulus feature, (2) a contextual rule, and (3) a temporal epoch.

Stimulus feature: choice decisions were based either on the letter (verbal task) or spatial location (spatial task). Participants pre-learned a sequence of letters (t-a-b-l-e-t) and locations (trace of a star, starting at the top position). Choice decisions were based upon these sequences.

Contextual rule: for a given stimulus feature, participants determined either whether the present stimulus was the first item of the relevant sequence ('t' for the verbal task; top position for the spatial task; sequence start task) or whether the present stimulus followed a reference stimulus in the sequence (sequence back task).

Temporal epoch: for the sequence back task, the reference stimulus was either the immediately preceding stimulus, or a stimulus that appeared multiple trials ago.

Loads on each of these factors were orthogonally varied to form a 2 × 2 × 2 factorial design. Trials were grouped into blocks wherein each block sampled one cell of the 2 × 2 × 2 design. In each block, the relevant stimulus feature, either letter or location, was cued by the color of the frame. The relevant stimulus feature remained constant throughout a block. Participants performed the sequence start task on the relevant stimulus feature on the first trial of a given block. Thereafter, each block was divided into three phases: a first baseline phase, a sub-task phase, and a second baseline phase. Transitions among these phases were indicated by the shape of the colored frames. In all blocks, the baseline phases were cued by square frames. Sub-tasks were cued by triangle, diamond, or cross frames with shape-to-sub-task mappings counter-balanced across participants. For ease of exposition in what follows, we will assume the following associations: circle-switching, triangle-planning, diamond-dual. Only one sub-task was cued in a given block.

In *Baseline* blocks, participants continued to perform the sequence back task throughout the block thereby keeping the contextual rule and temporal epoch constant. On *Switching* blocks, shape-switches (i.e. from square to circle or from circle to square) indicated the need to switch tasks (i.e. from sequence back to sequence start). Thus, these blocks placed demands on responding based upon the appropriate contextual rule engaging *contextual control*. On *Planning* blocks, triangle shapes indicated that the presented stimulus could be ignored. Participants acknowledged the presence of each triangle-framed stimulus with a 'no' keypress. All the while, the last square-shaped stimulus had to be retained as a reference for the next square-shaped stimulus. Hence, *Planning* blocks minimized processing of present stimuli, but placed demands on planning for the future in order to respond according to the correct temporal epoch thereby requiring *temporal control*. Finally, on *Dual* blocks, diamond frames indicated the need to both switch contextual rules, and plan for the future. That is, participants started and continued a new sequence on all diamond framed stimuli, but backwards-matched to the last square-framed stimulus when the frame reverted to a square. Hence, Dual blocks required both contextual and temporal control.

Different cognitive control process can be detailed through orthogonal contrasts of the factorial design. At the highest level, *temporal control* is isolated by contrasting blocks that require planning with those that do not (Dual + Planning > Switching + Baseline). At the middle level, *contextual control* is isolated by contrasting blocks that require switching with those that do not (Dual + Switching > Planning + Baseline). At the lowest level is *sensory-motor control* which can be examined through the contrast of (Dual + Baseline > Planning + Switching). To understand this contrast, consider that the Dual condition is effectively Switching + Planning. Then, using subtraction logic, what remains after subtraction is the Baseline condition which consists of the demands of selecting the appropriate feature and responding to it. The utility of using the contrast over-and-above simply examining the Baseline condition alone is that it better controls for ancillary demands, and it keeps all of the

main contrasts of interest statistically orthogonal to one another. Finally, those areas involved in sensory-motor control can be further fractionated by contrasting stimulus features with one another (i.e. verbal > spatial or spatial > verbal), thereby emphasizing verbal articulatory processes (verbal > spatial) or spatial attention processes (spatial > verbal).

Participants in sample 1 completed 24 blocks of each cell of the 2 × 2 × 2 design over the course of two fMRI sessions. These blocks consisted of 1920 total trials with each block containing 7–13 trials. Participants in sample 2 completed 12 blocks of each cell of the 2 × 2 × 2 design during a single fMRI session. These blocks consisted of 864 total trials with each block containing 7–11 trials. In both samples, the task was split across 6 runs of 16 blocks each in each session.

Each stimulus was presented for 500 ms followed by a variable inter-trial interval of 2600–3400 ms. At the end of each block, feedback indicating the number of correct trials in the block out of the total number of trials in the block was presented for 500 ms. A variable 2600–3400 ms interval separated each block. Self-paced breaks were administered in-between runs.

Within a week prior to scanning, participants performed a practice session to learn the task. During the practice session, participants received extensive written instruction and clarification from an experimenter. Given the numerous rules and complexities of the task, instruction was broken up such that participants first learned the verbal sequence, then the spatial sequence, and then the subtasks. Participants continued to repeat instruction and practice under experimental supervision until they were comfortable with the rules. Thereafter, participants completed three runs of the task on their own. During each scanning session, participants completed one additional practice run in the scanner prior to fMRI data collection.

## Cognitive battery

To obtain trait-level measures of cognitive ability, participants performed computerized versions of letter span, spatial span, operation span, and symmetry span. Operation span and symmetry span provide measures of working memory capacity (*Turner and Engle, 1989*). Letter span and spatial span provide simple measures of short-term memory capacity. Participants also performed a paper version of Raven's Advanced Progressive Matrices (http://www.perasonassessments.com), wherein Set I was used as practice, and Set II was used to measure fluid intelligence. As previously reported (*Nee and D'Esposito, 2016*), performance across these measures were correlated. Therefore, the measures were combined using principle components analysis with the first principle component providing an index of general higher-level cognitive ability.

## Image acquisition

As previously reported (*Nee and D'Esposito, 2016*; *Nee and D'Esposito, 2017*), brain imaging data were acquired at the Henry H. Wheeler, Jr. Brain Imaging Center at the University of California, Berkeley using a Siemens TIM/Trio 3T MRI equipped with a 32-channel head coil. Stimuli were projected to a coil-attached mirror using an MRI-compatible Avotec projector (Avotec, Inc; https://avotecinccom). Experimental tasks were created using E-Prime software version 2.0 (Psychology Software Tools, Inc; https://pstnet.com/). Eye position was monitored using an Avotec system (RE-5700) and Viewpoint software (http://www.arringtonresearch.com/). Response data were collected on an MR-compatible button box (Current Designs, Inc; https://curdes.com).

T2* weighted fMRI was performed using gradient echo planar imaging (EPI) with 3.4375 mm² in-plane resolution and 35 descending slices of 3.75 mm thickness. TR = 2000 ms, echo time = 25 ms, flip angle = 70, field of view = 220 mm². The first three images of each run were automatically discarded to allow for image stabilization. Field maps were collected to correct for magnetic distortion. High-resolution T1-weighted MPRAGE images were collected for anatomical localization and spatial normalization (240 × 256 × 160 matrix of 1 mm³ isotropic voxels; TR = 2300 ms; echo time = 2.98 ms; flip angle = 9). Each session included a 6 min eyes open resting state run in addition to six task runs. In sample 1, the resting-state run was collected prior to the task in the first session, and after the task in the second session. In sample 2, the resting-state run was collected prior to the task.

## Behavioral data preprocessing

Behavioral data was preprocessed using custom MATLAB code (MathWorks; https://www.mathworks.com). Data were sorted by phase (first baseline, sub-task, second baseline). Transitions

between phases were considered separately. Of primary importance in this report are the sub-task trials (i.e. all trials in the sub-task phase excluding the first trial of the phase) and what are referred to as 'return' trials which mark the transition from the sub-task phase to the second baseline phase. For example, in Planning blocks, the sub-task trials themselves require minimal processing since all stimuli during this phase require a 'no' keypress. On the return trial, participants perform sequence back with reference to the stimulus feature that preceded the sub-task phase. Thus, the sub-task trials reflect a combination of cognitive control in the moment and preparing for the future, while the return trials reflect actualization of sub-task preparation. Sub-task and return trials were separately sorted for each cell of the 2 × 2 × 2 design. Reaction times were computed on trials with a correct response only. Reaction times less than 200 ms were discarded as anticipatory and reaction times greater than 2000 ms were discarded as inattentive. Furthermore, reaction times greater than 2.5 standard deviations of the mean of a given condition were removed as outliers. These procedures resulted in the removal of 0.76% of the trials in sample 1, and 0.65% of the trials in sample 2.

## Image preprocessing

Unless otherwise specified, preprocessing was performed using SPM8 (http://www.fil.ion.ucl.ac.uk/spm/). Raw data was converted from DICOM into nifti format. Origins for all images were manually set to the anterior commissure. AFNI's 3dDespike function (http://afni.nimh.nih.gov/afni) was applied to the functional data to remove outlier time-points. Correction for differences in slice-timing and spatial realignment were performed next. Images were then corrected for magnetic distortion using the FieldMap toolbox implemented in SPM8 *Andersson et al., 2001*. Functional data were co-registered to the T1-weighted image. Segmented normalization was performed on the T1-weighted image. Spatial normalization parameters were subsequently applied to the functional task data. This step also resampled the functional images to 2 mm$^3$ isotropic. Task data were subsequently smoothed using an 8 mm$^3$ full-width/half-maximum isotropic Gaussian kernel. To ensure that smoothing did not artifactually influence the results, all analyses were also repeated with 4 mm$^3$ smoothing. Prior to spatial normalization, resting data underwent additional preprocessing using custom-written MATLAB scripts. Resting data were high-pass filtered at 0.009 Hz using SPM8's spm_filter function. Next, nuisance signals from white matter and cerebrospinal fluid were extracted and regressed out of the data along with linear, squared, differential, and squared differential motion parameters (motion parameters were estimated during spatial realignment described above). The residualized data were low-pass filtered at 0.08 Hz using a second-order Butterworth filter. The filtered, artifact-corrected resting data were then spatially normalized and smoothed as described above.

The bulk of the report focuses on data that were preprocessed in the manner described above. However, large volumetric smoothing kernels can blur and obscure activation gradients. To ensure that the observed activation gradients were not artifactually caused by these processing choices, the data were re-processed using Freesurfer version 6.0 (https://surfer.nmr.mgh.harvard.edu/). The T1-weighted image was registered to the fsaverage surface using recon-all. Despiked, slice-timing corrected, realigned, unwarped functional data described above (i.e. before spatial normalization) were sampled onto the fsaverage surface using preproc-sess (note, motion correction was omitted using the –nomc flag since motion correction was previously performed). Four mm smoothing was performed on the surface and then the functional data was returned to the volume using surfsmoothsess. In this case, returning the data to the volume enabled the same model estimation procedures to be performed on both the volume smoothed and surface smoothed data, providing a well-matched comparison.

## Univariate image analysis

Univariate modeling of task activation has been previously described (*Nee and D'Esposito, 2016*) and was implemented using SPM8. Of main interest are the sub-task phases which were separately modeled for each cell of the 2 × 2 × 2 design as epochs starting after the first sub-task trial through the end of the final sub-task trial. The first and second baseline trials were also modeled as epochs, separately for each phase and stimulus domain, also starting from the second trial of the epoch through the last. Transient regressors were included to model the first trial of each block, separately for each stimulus domain, the first trial of each sub-task, separately for each cell of the 2 × 2 × 2

design, and the first trial of the second baseline (also known as the 'return' trial), separately for each cell of the 2 × 2 × 2 design. Additional transient regressors were included to model left and right keypresses, as well as error trials. All regressors were convolved with the canonical hemodynamic response function implemented in SPM. Each run was modeled separately containing a separate intercept term, high-pass filter at 128 s, AR(1) modeling to account for temporal autocorrelation, and scaling such that run-wide global signal averaged 100. For participants demonstrating greater than 3 mm/degrees of motion over the course of the session or a single framewise movement greater than 0.5 mm/degrees, 24 motion regressors were included reflecting linear, squared, differential, and squared differential to remove motion-related artifacts (*Lund et al., 2005*; *Satterthwaite et al., 2013*).

At the first level, contrasts were created to isolate the sub-task phase of each cell of the 2 × 2 × 2 design by averaging parameter estimates across runs. These first level parameter estimates were then carried forward to a second level 2 × 2 × 2 ANOVA. Second-level contrasts were calculated as t-tests within the ANOVA framework. Five orthogonal contrasts of interest were considered: Temporal Control (Dual + Planning > Switching + Baseline), Contextual Control (Dual + Switching > Planning + Baseline), Sensory-Motor Control (Dual + Baseline > Planning + Switching), Verbal Stimulus Domain (Verbal > Spatial), and Spatial Stimulus Domain (Spatial > Verbal). Note that the contrasts with the 'Control' modifier collapsed across Stimulus Domain (e.g. Dual is a combination of Verbal Dual and Spatial Dual), while contrasts with the Stimulus Domain modifier collapsed across the Control conditions (e.g. Verbal is a combination of Verbal Dual, Verbal Planning, Verbal Switching, and Verbal Baseline). In keeping with previous reports on these datasets (*Nee and D'Esposito, 2016*; *Nee and D'Esposito, 2017*), whole-brain voxel-wise activation maps were thresholded at p<0.001 at the voxel level with 124 voxel extent providing correction for family-wise error (FWE) at p<0.05 according to AlphaSim.

## Network overlap

For each activation contrast, the overlap with the *Yeo et al., 2011*. 17-network parcellation was examined. For each of the 17 networks, the percentage of significant voxels for a given activation contrast that overlapped with the network was tabulated.

## Regions-of-interest

The five contrasts of interest produced wide swaths of activation throughout the PFC and PPC. To examine the activations in a more granular manner, regions-of-interest (ROIs) were created as 6 mm$^3$ spheres centered on peak activation coordinates. PFC ROIs have been previously described (*Nee and D'Esposito, 2016*; *Nee and D'Esposito, 2017*). Given that the contrasts produced partially overlapping activations, distinct peaks were defined as local maxima that were distanced at least 1.5 cm from another peak. The location of PFC peaks was determined by sample one and peaks similar in location revealed by the same contrast were identified in sample 2. This ordering was due to the temporal ordering of data collection and analysis (i.e. sample one was collected and analyzed first and sample two was analyzed in an effort to replicate findings in sample one as detailed previously [*Nee and D'Esposito, 2016*; *Nee and D'Esposito, 2017*]). This resulted in six PFC ROIs detailed in *Table 1*: lateral frontal pole (FPl), middle frontal gyrus (MFG), ventrolateral prefrontal cortex (VLPFC), caudal middle frontal gyrus (cMFG), inferior frontal junction (IFJ), and superior frontal sulcus (SFS).

ROIs in the PPC were defined separately for sample 1 and sample 2, again using the five contrasts-of-interest. Again, distinct peaks were identified as local maxima distanced from other peaks by at least 1.5 cm. This procedure results in four PPC ROIs detailed in *Table 1*: inferior parietal lobule (IPL), mid intra-parietal sulcus (midIPS), superior parietal lobule (SPL), and anterior intra-parietal sulcus (antIPS).

In both the PFC and PPC, some peaks could be identified by two or three contrasts (i.e. local maximum within 1.5 cm from one another were observed across multiple contrasts). The contrast chosen to define a given peak was done with consideration of the contrast that most unambiguously identified the peak in individual subjects. This choice was made since the effective connectivity analyses detailed below were tailored to each individual's activation peaks, providing individual-specific parcellation. The end result of this analysis choice was that the least expansive contrast (where

expansive refers to the number of activated voxels of a contrast) tended to be chosen to define an ROI for a given peak.

## Functional profiling and multi-dimensional scaling

Since ROIs could respond to multiple demands, we sought to quantify each ROI's activation profile. Because the ROIs were formed from peaks of a particular contrast in a particular sample, simply plotting the activation profile of the ROIs defined above constitutes circularity that biases the profile toward the ROI-defining contrast (*Kriegeskorte et al., 2009*). To provide unbiased estimates, the ROIs from sample one were used to functionally profile sample two, and vice versa. Hence, data used to define ROIs and test ROIs were independent. Within these unbiased ROIs, statistics for the five contrasts-of-interest were calculated using repeated-measures 2 × 2 × 2 ANOVAs. For each sample, statistical significance was Bonferroni corrected for the number of ROIs.

To further characterize each ROI, the ROI data were submitted to multi-dimensional scaling. The data for each individual and each ROI were z-scored across the eight conditions-of-interest. Then, for each ROI, the individual-level data were stacked, resulting in a two-dimensional matrix wherein one-dimension was composed of individuals x conditions, and the other dimension was composed of ROIs. Dissimilarity among ROIs was computed using $1 - r$, where $r$ is the Pearson's correlation. This dissimilarity matrix was submitted to MATLAB's cmdscale function. Each ROI was then plotted as a point in the space defined by the first two dimensions wherein the first two dimensions captured 89.2% of the variance in the data (71.0% for the first, 18.3% for the second, 0.04% for the third). As explained in the Results, the first dimension appeared to map well onto an abstraction axis, and the second dimension appeared to map well onto a stimulus domain axis.

## Brain-behavior relationships

To better understand how activation in each ROI relates to cognitive control, relationships between behavioral performance in the Comprehensive Control Task and activation were calculated similar to previous reports (*Nee and D'Esposito, 2016*; *Nee and D'Esposito, 2017*). Three differences between the analyses reported here and those reported previously is that here, (1) each ROI is considered separately (pairs of ROIs were combined previously), (2) ROI data were extracted to ensure independence as explained above (i.e. sample one defined ROIs for sample two, and vice versa), and (3) repeated measures correlations were computed using the rmcorr package in R (previously correlations were computed by regressing out subject-specific terms). Mean RT for the eight conditions of interest were calculated for each individual separately for the sub-task trials and return trials, as described above. The eight element vector was then z-scored separately for each phase. The sub-task trial RTs were regarded as 'Present Behavior' since the RTs reflect behavior at the time that the fMRI signal was estimated (i.e. activations of interest were drawn from the sub-task phase). The return trial RTs were regarded as 'Future Behavior' since the RTs reflect the time directly after the fMRI signal was estimated. Future Behavior reflects the behavior that is being prepared for during the sub-task phase, which might not be reflected in the sub-task trial RT per se (e.g. Planning blocks have fast Present RTs, but slow Future RTs). For each ROI, a repeated measures correlation was used to assess the relationship between activation and Present Behavior (controlling for Future Behavior), and activation and Future Behavior (controlling for Present Behavior). These procedures were performed separately for each sample. Statistical significance was assessed after Bonferroni correction for number of ROIs. Correlations that did not survive multiple comparisons correction, but were present at uncorrected levels are also noted.

Linear mixed effects models were fit in order to quantify how ROIs differ in terms of their relationships to Present and Future Behavior. For each individual and ROI, Present and Future Behavior were simultaneously regressed onto the activations. Parameter estimates reflecting the slope relating activation to Present and Future Behavior, respectively were submitted to the linear mixed effects models. For each ROI, the loading of the ROI on the first dimension of the multi-dimensional scaling described above was used as a continuous, objective measure of abstraction. Separate linear mixed effects models tested whether the relationship between Present RT and activation varied as a function abstraction, and whether the relationship between Future RT and activation varied as a function of abstraction. That is, these analyses tested whether the slope estimates varied by abstraction. Models of the form: Present Behavior Slope ~ Abstraction + (Abstraction|Subject), and Future

Behavior Slope ~ Abstraction + (Abstraction|Subject) were estimated. These analyses were performed separately for each sample.

ROI analyses offer an incomplete picture of brain-behavior relationships. To better visualize whole-brain patterns, voxel-wise partial correlations were performed. The data were stacked across individuals. For each voxel, the partial correlation between activation and Present Behavior, controlling for Future Behavior and individual, as well as the partial correlation between activation and Future Behavior, controlling for Present Behavior and individual were calculated. The resultant maps were thresholded at p<0.001 at the voxel-level, and clusters of 124 voxels or more were retained for visualization.

## Resting-state functional connectivity

Of main interest in this report is effective connectivity which estimates directed relationships among areas. To constrain these analyses, seed-based functional connectivity was performed on the resting-state data in order to identify ROI's that may be connected to one another (and prune connections that are likely to not be present). For each ROI, voxel-wise correlations were computed between the ROI time-series and each voxel time-series. This was done separately for each resting-state run in sample 1 (note, only sample 1 was used here to maintain compatibility with the connectivity matrix defined previously [*Nee and D'Esposito, 2016*]). For a given seed ROI, the correlation values of each voxel were ranked after excluding voxels within 1.5 cm of the seed ROI. A correlation value $r$ at cost $c$ was computed wherein $r$ reflects the correlation value of voxel at the $c$th percentile. Here, $c$ was set to 18 as detailed previously (*Nee and D'Esposito, 2016*). Then, four counts were made: whether ROI 1 was connected to ROI 2 when ROI 1 was the seed in run 1, whether ROI 1 was connected to ROI 2 when ROI 2 was the seed in run 1, whether ROI 1 was connected to ROI 2 when ROI 2 was the seed in run 2, and whether ROI 1 was connected to ROI 2 when ROI 2 was the seed in run 2. A pair of ROIs were considered 'connected' if the average correlation value in the target ROI exceeded $r$. If at least 3 of the 4 counts were tallied, the connection between the areas was considered a candidate for future consideration. The resultant candidate connectivity matrix was utilized for effective connectivity analyses. The intent of this procedure was to be somewhat liberal, and allows the effective connectivity modeling to further prune connections. Hence, this procedure should be considered a scaffold for analyses to follow rather than an attempt to determine the true connectivity structure of the data.

## Spectral dynamic causal modeling

Time-invariant effective connectivity was estimated using spectral dynamic causal modeling (spDCM) implemented in SPM12 and DCM12.5. spDCM inverts a biophysically plausible generative model of the fMRI cross-spectral density (*Friston et al., 2014*; *Razi et al., 2015*; *Razi et al., 2017*). The model estimates the effective connectivity among hidden neural states and includes a hemodynamic forward model that accounts for how synaptic activity translates into regional hemodynamics and observed connectivity in the fMRI signal. spDCM estimated the effective connectivity among the six PFC and four PPC areas that are the focus of this report. Previously, 'classic' deterministic dynamic causal modeling (DCM) (*Friston et al., 2003*) was performed on the six PFC areas (*Nee and D'Esposito, 2016*; *Nee and D'Esposito, 2017*). 'Classic' DCM differs in numerous algorithmic and practical ways from spDCM (see *Friston et al., 2014*). Most notably, while 'classic' DCM inverts a generative model of the time-series of each modeled area, spDCM inverts a generative model of the cross-spectral density. The latter renders spDCM much more computationally tractable. This computational tractability comes at the cost of limiting modeling to time-invariant aspects of effective connectivity, which differs from 'classic' DCM that models both time-invariant and time-varying aspects of effective connectivity. However, 'classic' DCM becomes computationally impractical as the number of modeled areas grows. Therefore, spDCM is the more appropriate approach given the number of areas considered here. To complement spDCM, psycho-physiogical interaction (PPI) analyses were employed (described below) to capture time-varying aspects of effective connectivity. Hence, relative to previous work (*Nee and D'Esposito, 2016*; *Nee and D'Esposito, 2017*), the present study employed different methods that are computationally better-suited to answering questions at the network-level. An added benefit of its computational tractability is that the present approach

can also scale up to answering questions about the interactions of the FPCN and other networks, which will facilitate future work.

Given that each individual has a specific topography in the PFC (*Braga and Buckner, 2017*), volumes of interest (VOIs) were defined based on each individual's activation peaks. A 6 mm$^3$ sphere was created around each individual's activation peak that was close to the peak in the group contrast. To reduce noise in the VOIs, tiered thresholding was performed to eliminate unresponsive voxels. This threshold was started at p<0.001 (80.0% of VOIs), was lowered to p<0.05 if voxels did not survive (14.4% VOIs). If voxels still did not survive, the VOI was centered at the group peak and the threshold was lowered further to p<0.5 (5.1% of VOIs), and the threshold was eliminated if this was still insufficient (0.004% of VOIs). The model architecture utilized the candidate connectivity matrix estimated from the resting-state data described above.

spDCM's required the estimation of 82 parameters using the model architecture based upon the candidate connectivity matrix. To provide sufficient data to estimate these parameters, pairs of runs were collapsed together using a modified version of spm_fmri_concatenate. The modifications allowed a flexible number of runs to be concatenated (default behavior is to concatenate all runs), and added a regressor to account for edge effects caused by run-wise (i.e. not concatenated) high-pass filtering. Other changes relative to the univariate modeling was that models of all participants included linear, squared, differential, and squared differential motion parameters, as well as a regressor to capture framewise displacement calculated across the six motion parameters (*Power et al., 2012*). The time-series' for every VOI was extracted after regressing out all regressors related to confounds of non-interest (i.e. run means, motion, filter edge effects).

For each participant, two spDCM's were estimated. The first modeled effective connectivity on time-series' that excluded only confounds signals. The second modeled effective connectivity on time-series' that additionally regressed out all task-related regressors included in the univariate modeling. Although spDCM is designed to estimate time-invariant effective connectivity, it is possible that task-related activity may nevertheless influence effective connectivity estimates (*Cole et al., 2019*). However, each of these spDCM's produced similar results with no evidence of bias produced by including/excluding task-related regressors (*Figure 6—figure supplement 1*) consistent with the idea that spDCM estimated time-invariant effective connectivity as designed.

The following defaults were changed: the order of the AR model was set to 4 (this was the default for DCM12, but the default is 8 in DCM12.5). This was done after observing systematically poorer fits at the default levels in preliminary analysis. The maximum number of nodes to model was increased to 10 to accommodate the number of areas of interest. Finally, the maximum number of iterations for model fitting was increased to 1000 to ensure that all models converged.

spDCM was performed separately for each sample. This enabled the examination of the replicability of effective connectivity estimates and derivations thereof.

## Psychophysiological interaction analysis

To examine dynamic (time-varying) effective connectivity, psychophysiological interaction (PPI) analysis was performed. Here, a variant of the method detailed in *Cole et al., 2013* was employed. Time-series' from the same VOIs that were submitted to spDCM were modeled (in this case, the VOIs without the task regressed out since the task regressors are regressed out as part of PPI). For each connection in the probable connectivity matrix, the data were modeled as:

$$Y_{target} = B_{conn}Y_{source} + B_{uni}X + B_{dep}(\text{bin}(X)^*C.^*Y_{source})$$

Here, $Y_{target}$ is the time-series of the target VOI, $Y_{source}$ is the time-series of the source VOI, X is the univariate design matrix, bin(X) is a binarized version of the univariate design matrix wherein time-points that are above 0 are set to one and other time-points are set to 0, and C is the contrast matrix. This results in estimations of several contributions to the signal in the target VOI: $B_{conn}$ which represents the time-invariant functional connectivity among the target and source, $B_{uni}$ which represents the univariate time-invariant task signals, and of main interest, $B_{dep}$ which is the context-dependent changes in effective connectivity from the source to the target. This formulation is identical to *Cole et al., 2013* except that contrasts are included within the model itself rather than being computed after parameter estimation. This change was performed because it reduces multi-collinearity, which is particularly problematic for PPI among highly correlated nodes. Preliminary analyses

revealed that the more connected two regions were, the more negatively biased $B_{dep}$ became using the method of *Cole et al., 2013*. This bias was eliminated by utilizing contrasts within the model.

Context-dependent connectivity (i.e. the PPI) is dictated by the contrasts defined in C. Here, five contrasts were defined which correspond to the four univariate contrasts of interest (Temporal Control, Contextual Control, Sensory-motor Control, Stimulus Domain) plus an additional interaction contrast that modeled Contextual Control x Stimulus Domain.

PPI analyses were performed separately for each sample. This enabled the examination of the replicability of the effective connectivity estimates and derivations thereof. PPI's of Sensory-motor Control and Contextual Control x Stimulus Domain were unreliable across samples (*Figure 7—figure supplement 1*). Therefore, analyses were restricted to PPI's of Temporal Control, Contextual Control, and Stimulus Domain.

## Predicting higher level cognitive ability

spDCM and PPI provided measures of static and dynamic effective connectivity, respectively. In particular, spDCM analyses revealed a role of contextual control-related areas in integrating cognitive control networks, and PPI analyses revealed greater integration across cognitive control networks during contextual control conditions. To examine the relevance of these measures to higher level cognition, static and dynamic effective connectivity were used to predict individual-differences in general higher-level cognitive ability.

General higher-level cognitive ability was assessed using the first principle component of the scores on the cognitive battery described above. To provide an index of static integration, for each network, between-network effective connectivity was contrasted against within-network effective connectivity. To isolate static integration of the contextual control network, the residualized static integration of the contextual control network was utilized after regressing out the static integration indices of the temporal control and sensory-motor control networks, as well as the root-mean square of framewise displacement (RMS FD) as a confound. Dynamic integration was calculated in a parallel manner to static integration. In this case, between-network effective connectivity was contrasted with within-network effective connectivity across all networks, but isolated to particular contrasts. For example, dynamic integration of contextual control indexed the degree to which there was greater between- relative to within-network communication for the contextual control contrast. Intuitively, whereas static integration of the contextual control network indicates the potential for contextual control areas to integrate networks, dynamic integration during contextual control indicates the actualization of this integration during the conditions that maximally activate contextual control areas. Similar to the static integration index, residualized dynamic integration during contextual control was utilized after regressing out the dynamic integration indices during temporal control and stimulus processing, along with RMS FD as a confound. To ensure independence among our samples, each of these indices (general higher level cognitive ability, static integration, dynamic integration) were calculated separately on sample 1 and sample 2.

Ridge regression was performed to predict general higher-level cognitive ability from static and dynamic integration. This was done in a two-fold manner, wherein each fold divided the dataset into training and test sets as a function of the samples (i.e. sample 1 and sample 2). For example, sample one was used to estimate weights using ridge regression, and then those weights were used to predict general higher-level cognitive ability in sample 2. Ridge regression was performed with at an arbitrarily chosen penalty parameter = 1, although significant prediction was achieved at penalties as high as 100 (which was the highest tested). The correlation between the observed and predicted general higher-level cognitive abilities was used to assess prediction performance. The observed correlation was compared to a permutation distribution formed by randomly permuting the rows of the prediction matrix 1000 times.

## Predicting transcranial magnetic stimulation susceptibility

As previously reported (*Nee and D'Esposito, 2017*), sample two was used to test the causal role of frontal areas to different forms of cognitive control. Each participant in sample two underwent an additional four sessions of the Comprehensive Control Task following continuous theta-burst stimulation (cTBS) to each of FPl, VLPFC, SFS, and S1 (control site). cTBS is patterned form of transcranial magnetic stimulation (TMS) thought to inhibit cortical tissue (*Huang et al., 2005*). Each TMS target

was a node in a distinct PFC-PPC network and was predicted to produce distinct impairments in cognitive control. To examine whether static and dynamic integration of cognitive control networks is related to TMS susceptibility, the three observed TMS effects of interest were combined: the effect of FPl TMS on temporal control on error-rate, the effect of VLPFC TMS on the contextual control x stimulus-domain interaction on reaction time, and the effect of SFS TMS on stimulus-domain on error-rate (see *Nee and D'Esposito, 2017* for additional details). For each effect, the frontal TMS effect was isolated by regressing out the corresponding effect following S1 TMS. Then, all TMS effects were combined using the first principle component.

Leave-1-out ridge regression was performed to predict the general cognitive control TMS effect from static and dynamic integration, wherein static and dynamic integration were computed identically to the above. The correlation between the observed and predicted general higher level cognitive abilities was used to assess prediction performance. The observed correlation was compared to a permutation distribution formed by randomly permuting the rows of the prediction matrix 1000 times.

## Acknowledgements

This research was supported by National Institute of Neurological Disorders and Stroke Grants F32 NS0802069 (DN) and P01 NS040813 (Mark D'Esposito), National Institute of Mental Health Grants R01 MH063901 (Mark D'Esposito) and R01 MH121509 (DN), and Florida State University COFRS Award 0000034175 (DN). The author thanks Regina Lapate, Mac Shine, and Anila D'Mello for helpful discussions of this work.

## Additional information

### Funding

| Funder | Grant reference number | Author |
| --- | --- | --- |
| National Institute of Neurological Disorders and Stroke | F32 NS0802069 | Derek Evan Nee |
| National Institute of Mental Health | R01 MH121509 | Derek Evan Nee |
| Florida State University | COFRS 0000034175 | Derek Evan Nee |
| National Institute of Neurological Disorders and Stroke | P01 NS040813 | Derek Evan Nee |
| National Institute of Mental Health | R01 MH063901 | Derek Evan Nee |

The funders had no role in study design, data collection and interpretation, or the decision to submit the work for publication.

### Author contributions

Derek Evan Nee, Conceptualization, Data curation, Software, Formal analysis, Funding acquisition, Validation, Investigation, Visualization, Methodology, Writing - original draft, Writing - review and editing

### Author ORCIDs

Derek Evan Nee https://orcid.org/0000-0001-7858-6871

### Ethics

Human subjects: Informed consent was obtained in accordance with the Committee for Protection of Human Subjects at the University of California, Berkeley (2010-04-1314, 2010-02-781).

### Decision letter and Author response

Decision letter https://doi.org/10.7554/eLife.57244.sa1

Author response https://doi.org/10.7554/eLife.57244.sa2

## Additional files

### Supplementary files

• Supplementary file 1. Tables containing supplemental behavioral analyses. Factorial ANOVA's for the sub-task trials and return trials. RT – reaction time; ACC – accuracy; ME – main effect.

• Transparent reporting form

### Data availability

All data and code needed to reproduce the figures in this report can be found at https://osf.io/938dx/.

The following dataset was generated:

| Author(s) | Year | Dataset title | Dataset URL | Database and Identifier |
|---|---|---|---|---|
| Nee D | 2020 | PFC-PPC Integration | https://osf.io/938dx/ | Open Science Framework, 938dx |

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

# Appendix 1

## Supplemental results

### Supplemental behavioral results

Behavioral data have been previously reported (*Nee and D'Esposito, 2017*; *Nee and D'Esposito, 2016*). Here, the focus is on two phases of the data: the sub-task phase wherein cognitive control demands are manipulated, and the return trials, wherein preparatory control processes readied during the sub-task phase are expressed.

Due to the complex design, it is useful to highlight particular data patterns that provide assurance that the task manipulations operate as desired. The interested reader is directed to *Supplementary file 1* for complete factorial statistics.

First, effects of stimulus domain were examined. In sample 1, RTs on verbal and spatial tasks were well-matched during the sub-task (verbal: 644.32 ms; spatial: 642.4 ms; $F(1,23) = 0.03$, p=0.87) and return trials (verbal: 829.5 ms; spatial: 823.7 ms; $F(1,23) = 0.12$, p=0.74). Accuracies were also similar during sub-task (verbal: 92.8%; spatial: 93.9%; $F(1,23) = 3.12$, p=0.09), and return trials (verbal: 0.79%; spatial: 0.81%; $F(1,23) = 3.01$, p=0.10), though trending towards more errors on the verbal task. In sample 2, RT data provided some evidence that the verbal task was more difficult than the spatial task on sub-task trials (verbal: 792.9 ms; spatial: 757.8 ms; $F(1,24) = 5.01$, p=0.03), with a similar trend on return trials (verbal: 932.7 ms; spatial: 908.0 ms; $F(1,24) = 3.80$, p=0.06). Accuracies were not significantly different on sub-task (verbal: 94.5%; spatial: 95.4%; $F(1,24) = 2.90$, p=0.10) nor return trials (verbal: 84.5%; spatial: 84.6%; $F(1,24) = 0.01$, p=0.91). Combining the samples, there was no significant difference in verbal and spatial RTs during the sub-task ($F(1,48) = 3.55$, p=0.07) and return trials ($F(1,48) = 2.32$, p=0.13). There was a significant difference in verbal and spatial accuracies during the sub-task ($F(1,48) = 6.09$, p=0.02), but not return trials ($F(1,48) = 1.62$, p=0.21). Collectively, these data provide some scattered evidence that the verbal task was modestly more challenging than the spatial task, but overall, the stimulus domains were reasonably well-matched.

Second, it is desirable that the cognitive control manipulations produce clear and consistent effects on behavior. To illuminate these effects, I consider simple comparisons to the *Baseline* condition, although clear and consistent effects are observed using the full factorial framework and detailed in Supplemental Tables 1 and 2. *Sensory-motor control* involves selecting an appropriate action based upon the stimulus. Demands on these processes are minimized during the *Planning* sub-task phase, during which participants can reflexively respond without processing the verbal or spatial stimulus features. Correspondingly, RTs are expected to be speeded during the *Planning* sub-task phase. These expectations were met in both samples (sample 1: mean difference = −184.1 ms; Cohen's d = 2.09; $t(23) = −10.24$, p=4.85e-10; sample 2: mean difference = −211.7 ms; Cohen's d = 1.61; $t(24) = −8.06$, p=2.75e-8). Accuracies were also elevated in the Planning sub-task phase relative to Baseline indicating that RT speeding did not tradeoff from accuracy (sample 1: mean difference = 5.3%; Cohen's d = 0.87; $t(23) = 4.25$, p=0.0003; sample 2: mean difference = 3.6%; Cohen's d = 0.71; $t(24) = 3.56$, p=0.002).

Next, *contextual control* involves selecting the appropriate task set over-and-above sensory-motor control. This should incur an additional cost in RT reflected when comparing the *Switching* sub-task phase to the *Baseline* condition. These expectations were met in both samples (sample 1: mean difference = 60.8 ms; Cohen's d = 1.08; $t(23) = 5.31$, p=2.17e-5; sample 2: 189.0 ms; Cohen's d = 1.53; $t(24) = 7.65$, p=6.90e-8). Accuracy was either the same or slightly worse in the Switching condition, indicating that slowing was not a result of speed-accuracy tradeoff; sample 1: mean difference = 0.4%; Cohen's d = 0.10; $t(23) = 0.51$, p=0.62; sample 2: mean difference = −1.8%; Cohen's d = −0.43; $t(24) = −2.16$, p=0.04.

Finally, *temporal control* involves utilizing temporally extended representations to guide action. In the *Planning* condition, these representations are sustained during the sub-task phase, and utilized during the return trial. Hence, demands on temporal control are most clearly manifest in behavior on the return trial. In contrast to the above analyses that showed speeded RTs during the sub-task phase of the Planning condition relative to the Baseline condition, this same comparison on return trials showed slowed RTs for the Planning condition in both samples (sample 1: mean difference = 88.8 ms; Cohen's d = 0.65; $t(23) = 3.20$, p=0.004; sample 2: mean difference = 71.9 ms; Cohen's d = 0.59; $t(24) = 2.96$, p=0.007). Accuracies were also diminished (sample 1: mean

difference = −13.1%; Cohen's d = 0.93; t(23) = −4.54, p=0.0001; sample 2: mean difference = −6.5%; Cohen's d = 0.63; t(24) = −3.16, p=0.004).

Collectively, the behavioral data produced the expected cognitive control-related patterns.

## Supplemental imaging results

Although the analyses in the main text suggest that integration predicts a general susceptibility of cognitive control to TMS, it leaves open whether such effects are domain-general or domain-specific. The first principle component of the TMS effects loaded relatively equally on all TMS effects examined (0.67 on effect of FPl-TMS to temporal control; 0.49 on effect of SFS-TMS to spatial stimulus-domain; 0.56 on effect of VLPFC-TMS to verbal contextual control). The second principle component of TMS effects loaded positively on spatial effects, negatively on verbal effects, and negligibly on domain-general effects (−0.07 on effect of FPl-TMS to temporal control; 0.79 on effect of SFS-TMS to spatial stimulus-domain; −0.62 on effect of VLPFC-TMS to verbal contextual control) suggesting that the second principle component reflected domain-specific effects of TMS (i.e. spatial > verbal). Whereas static and dynamic integration were able to predict the first principle component of TMS effects as described in the main text, the second component reflecting domain-specific effects could not be predicted using analogous methods (r = 0.03, p>0.3). These data indicate that the predictive power of the integration measures are limited to domain-general cognitive control.

