## [Decision Letter]

**Acceptance summary:**

The authors investigated how functional gradients of cognitive control might manifest in the fronto-parietal control network. The results suggest that the gradients in this network might reflect changes between internally-oriented vs. externally oriented control processes involved in planning for the future and specifying actions according to the present stimuli, respectively. An intermediary zone in this network might play an integrative function by providing the necessary contextual control.

**Decision letter after peer review:**

Thank you for submitting your article "Integrative frontal-parietal dynamics supporting cognitive control" for consideration by *eLife*. Your article has been reviewed by Timothy Behrens as the Senior Editor, a Reviewing Editor, and three reviewers. The following individuals involved in review of your submission have agreed to reveal their identity: Evan Gordon (Reviewer #2); Moataz Assem (Reviewer #3).

The reviewers have discussed the reviews with one another and the Reviewing Editor has drafted this decision to help you prepare a revised submission.

Summary:

Nee presents a sophisticated set of new analyses on two previously published fMRI/TMS data sets to further examine the proposal of functional gradients of cognitive control in lateral prefrontal cortex (PFC) and posterior parietal cortex (PPC). The two prior papers have produced evidence for an integrative "apex" of control in the mid-lateral PFC. The current paper (a) provides a reconceptualization of these gradients as reflecting an externally to internally focused dimension (and offers some associated fMRI analyses), and (b) applies a series of connectivity and brain-behavior correlation analyses to test the integrative role of mid-lateral PFC and PPC regions. In particular, the author demonstrated that the Fronto-Parietal control network has a rostro-caudal organization, in which rostral regions are more engaged during sensory-motor control, caudal regions are more engaged during temporal control, and central regions are engaged during both; and further, activation of these regions predict behavior in a domain-specific fashion. He provides evidence that the central "Context Control" regions are the regions that are exerting primary influence over the other regions, both statically and especially dynamically, and that these static and dynamic influences are associated with both baseline cognitive abilities and with susceptibility to TMS effects. An understanding of the neuroanatomy of high-level control processes is a topic of great importance, and the basic approach (using a very complex and clever task) and analyses pursued by the author are compelling and sophisticated, and they ultimately produce a pretty coherent set of results.

Essential revisions:

1) The author should explain more clearly what aspects of the current paper are new. A devil's advocate (more precisely, a "novelty advocate") could argue that these functional gradients in PFC have been reported previously (in Nee and D'Esposito, 2016, 2017), and the same seems to be true for the PPC (Choi et al., 2018). Moreover, the proposal that the gradients that have previously been cast in terms of levels of abstraction (Badre) or temporal information integration (Koechlin) could also be thought of as reflecting an external-to-internal orientation of cognition is interesting but could be argued to just shift the semantics a little bit, without really widening the scope of phenomena explained. Similarly, some of the current "new" analyses could be argued to either reiterate previous results (Figure 2), re-package the previously reported results in slightly different ways (e.g., the MDS results in Figure 3), or represent foregone conclusions (the "present vs. future" results in Figure 4 seem to follow pretty inevitably from the way the task and brain regions' roles are being re-conceptualized, if I am not mistaken). That would leave only the connectivity-based analyses as providing new information, and even here it is not clear what is known already and what is novel (as DCM was already applied to the PFC data in the 2016 paper). Altogether, this created a murky impression as to what is old, what is new-ish (but fully expected within the re-conceptualized framework), and what is actually novel (in a nontrivial sense) in the results.

2) While the author frames the cortical subdivisions observed here as subcomponents of the Fronto-parietal control network, it is not entirely convincing that this is an accurate description of these areas. The "temporal control" areas, in localizing to angular gurus, far anterior middle frontal gyrus, and anterior temporal cortex, could arguably be better described as being within the Default network. The "Sensory-motor control" regions, in localizing to regions such as SPL and SFS, could be better described as being within the Dorsal Attention network. Certainly, the described differential functions of these regions fit well with known functions of these two networks. This could be adjudicated by a) showing medial surface views of Figure 2 (helpful especially for determining whether Default regions are involved), and b) computing an overlap metric of each set of regions (or activation map) with a publicly available network map, such as that described in (Yeo et al., 2011). If these regions do overlap better with non-FP networks, this doesn't necessarily invalidate any of the paper's findings, but it does change their interpretation somewhat: instead of reasoning that a central region of FP network regulates peripheral FP regions, we might conclude that FP regulates other networks, in a fashion similar to that outlined in (Gratton et al., 2018).

3) While the author has framed these results in the context of a macroscale gradient in frontal (and parietal) cortex, there is no strong evidence for this. Three sets of regions with different functions does not necessarily indicate a gradient; it could indicate three discrete networks. To make strong claims about finding a gradient, the author would need to test a continuous line of ROIs from one of the current seeds to the next, and show that brain function varies continuously with position. And ideally this would need to be done in individual subjects, to avoid the blurring that comes from averaging across subjects with variable spatial organization of FP (as detailed in Braga and Buckner, 2017). When such tests have previously been done (in resting-state data, at least), they indicate the existence of strong borders between discrete areas in lateral frontal cortex, not continuous gradients (see e.g. Wig et al., 2014, esp. Figure 7). It would not be completely necessary to perform such a test, because this concern doesn't invalidate any of the findings here. However, absent more comprehensive tests, the gradient interpretation might be simply only one of several frameworks that should be discussed. Towards that end, the author could make better contact with previous works that have subdivided the brain into discrete sub-networks (rather than framing it as a continuous gradient) (Abou Elseoud et al., 2011; Doucet et al., 2011; Meunier et al., 2009), and especially those that have characterized how FP subdivisions connect strongly with other networks. (Gordon et al., 2018) and particularly (Dixon et al., 2018) seem highly relevant here, and they fit very nicely with the present characterization of the central FP network providing separate top-down integrative control of internal and external stimuli.

4) The results in this manuscript leave open the question of how they can be reconciled with strong evidence that multitudes of tasks co-activate highly specific regions within the fronto-parietal cortices that largely overlap with regions of the intermediate zone (e.g. Shine et al., 2016; Fedorenko et al., 2013; Assem et al., 2020). Importantly, a recent study using hundreds of subjects and improved brain alignment methods (Assem et al., 2020) showed that even a simple 0-back task (similar to the baseline task used in this study) co-activates both caudal and rostral fronto-parietal regions. Is it possible the current task contrasts are missing out on regions with signals of comparable strengths? For example, in the sensory-motor contrast (Dual + baseline > Planning + switching), could the baseline task be also engaging broader frontal and parietal regions though more weakly than the other tasks? This would suggest that the intermediate zone has relative rather than absolute concrete/abstract gradient preferences.

5) An important methodological concern is that all the analysis used data with excessive unconstrained volumetric smoothing (8 mm) mixing signals from functionally distinct regions (see Coalson et al., 2018). Finding replicable results across two independent samples analyzed using the same non-optimal methods does not rule out the possibility of falling in the same pitfalls in both datasets. For example, while the manuscript claims to examine gradients within the fronto-parietal network, the temporal control contrast might be mainly highlighting DMN activations and connectivity dynamics. This also limits the interpretability of the connectivity indices and their relation to individual differences (Bijsterbosch et al., 2018, 2019). Ideally, the author is encouraged to repeat the task and connectivity analyses with minimal smoothing (2-4 mm). Further, why are all the analyses limited to the left hemisphere? If it is because the task effects are stronger on the left, this does not exclude decent effects on the right hemisphere which could provide a strong validation for the task and connectivity results.

6) To improve the interpretability of the effective connectivity results, it would be helpful to provide more details on the methodological and biological assumptions behind using positive and negative effective connectivity indices to suggest that they reflect excitation and inhibition, respectively, as well as the limitations related to fMRI effective connectivity analyses. Further, negative correlations between static connectivity and general cognitive abilities are in contrast with numerous reports showing that measures of resting-state functional connectivity across the fronto-parietal network positively correlate with general cognitive abilities (e.g. Dubois et al., 2018). Could the author comment on this difference?

Also, the author interprets the negative between-network connectivity as reflecting functional segregation, and positive connectivity as reflecting integration. This is plausible but certainly not the only possible interpretation of such a pattern. For instance, increases in between-node connectivity with greater control demand are often interpreted as reflecting modulation/biasing of one region by the other, which seems conceptually distinct from "integration", at least the way the author conceptualizes integration (i.e., the mid-lateral PFC joining up information from posterior and anterior regions). I wonder whether there might be stronger tests of the claim that this node integrates information processed in the other nodes, perhaps by using an information-based analytic approach (like MVPA/RSA)?

7) It would be problematic at brain-behavior individual differences using a small sample size like the one used to probe connectivity-TMS effects due to their unreliability. If the author insists on keeping this section, it will be useful to examine any shared variance between general cognitive abilities and the observed TMS behavioral effects.

[Editors' note: further revisions were suggested prior to acceptance, as described below.]

Thank you for resubmitting your work entitled "Integrative frontal-parietal dynamics supporting cognitive control" for further consideration by *eLife*. Your revised article has been evaluated by Timothy Behrens (Senior Editor) and a Reviewing Editor.

Summary:

The authors investigated how functional gradients of cognitive control might manifest in the fronto-parietal control network. The results suggest that the gradients in this network might reflect changes between internally-oriented vs. externally oriented control processes that are involved in planning for the future and specifying actions according to the present stimuli. An intermediary zone in this network might play an integrative function by providing the necessary contextual control.

The manuscript has been improved but there are some remaining issues that need to be addressed before acceptance, as outlined below:

1) In the Discussion section "Beyond the PFC: Towards a Network View of Cognitive Control" the author gives the impression that the majority of studies on cognitive control have solely focused on the PFC. This needs to be rebalanced to reflect that a distributed whole brain network view of cognitive control has long been argued (to name a few: Cole and Schnider, 2007; Power and Petersen, 2013; Fedorenko et al., 2013; Warren et al., 2014; Duncan et al., 2010 and 2020).

2) Some of the supplementary figures were missing (Figure 2—figure supplement 2, Figure 6—figure supplement 2, Figure 7—figure supplement 3), not sure if this was a problem with the submission system. There were also a few typos.

---

## [Author Response]

Essential revisions:1) The author should explain more clearly what aspects of the current paper are new. A devil's advocate (more precisely, a "novelty advocate") could argue that these functional gradients in PFC have been reported previously (in Nee and D'Esposito, 2016, 2017), and the same seems to be true for the PPC (Choi et al., 2018). Moreover, the proposal that the gradients that have previously been cast in terms of levels of abstraction (Badre) or temporal information integration (Koechlin) could also be thought of as reflecting an external-to-internal orientation of cognition is interesting but could be argued to just shift the semantics a little bit, without really widening the scope of phenomena explained. Similarly, some of the current "new" analyses could be argued to either reiterate previous results (Figure 2), re-package the previously reported results in slightly different ways (e.g., the MDS results in Figure 3), or represent foregone conclusions (the "present vs. future" results in Figure 4 seem to follow pretty inevitably from the way the task and brain regions' roles are being re-conceptualized, if I am not mistaken). That would leave only the connectivity-based analyses as providing new information, and even here it is not clear what is known already and what is novel (as DCM was already applied to the PFC data in the 2016 paper). Altogether, this created a murky impression as to what is old, what is new-ish (but fully expected within the re-conceptualized framework), and what is actually novel (in a nontrivial sense) in the results.

I acknowledge this perspective, especially given the way in which the manuscript was originally laid out. As I explain below, I have made edits to help guide the reader towards what I take as the main novelty of the manuscript (the connectivity-based analyses), indicate more clearly what might be considered more incremental based on prior work (functional analyses), and also explain that what sets this work apart is the combination of these two approaches which I take as greater than the sum of their parts. I unpack the reasoning behind this below:

Much of the novelty is in the connectivity-based analyses as has been pointed out (see more below). The intention of the analyses that precede the connectivity-based analyses are to provide a functional foundation from which the connectivity-based analyses can be interpreted. That is, showing that network A and network B interact in manner X has theoretical impact only insofar as one can determine, to a reasonable approximation, what functions networks A and B perform. In contemporary cognitive neuroscience, these functional underpinnings are oftentimes weakly and/or coarsely inferred. Consider, for example, that what I refer to as the “temporal control” network was interpreted in the previous round of reviews as the “default-mode” network (see Essential revision 2). Below I address this issue by computing overlaps with a widely used atlas, but I could argue that more direct evidence against such an interpretation can be levied by careful inspection of Figure 3 and Figure 4. In Figure 3, one can observe that areas within the temporal control network are more strongly activated in both the most demanding (dual) and least demanding (planning) conditions relative to the baseline condition. Such a pattern does not seem to accord with the (negative) monotonic demand-activation relationship typically observed in “default-mode” areas. Moreover, although a labored argument might be constructed to explain how “default”-related activations relate to future behavior (defined as reaction times in trials immediately following the sub-task period), the most straightforward explanation of this pattern is that these activations reflect preparation for the future (a control process), rather than some off-task internal mentation (a “default” process). Hence, the rich functional data offer detailed insights that significantly exceed what could be gained from a functionally unconstrained method such as resting-state, or a functionally coarse method such as quantitative reverse inference and/or a task battery. This precise characterization provides a firm functional foundation from which to interpret network interactions. Therefore, much of the manuscript is devoted to establishing that foundation, and I take it as an important innovation of the present manuscript.

In my view, properly establishing a functional foundation required some re-depicting of analyses previously described (PFC ROI analyses), demonstrating parallels in the PPC (replicating Choi et al., 2018), expanding upon the parallels in the PPC (demonstrating timescale/medium-specific brain-behavior relationships) and providing a theoretical perspective to synthesize the results (external/present-to-internal/future axis). It can be argued that the knowledge gained in this section of the manuscript is more incremental than novel, but without this foundation, I feared that the reader would be unable to interpret the network-based analyses. I do share the concern that perhaps the true novelty of the manuscript, which I take to be the network-based analyses, comes too late and a reader might be led to believe that the functional analyses are the focus. One possible solution to this issue would be to push some parts of Figure 2, Figure 3 and Figure 4 to supplemental material. I attempted this in a draft but was unsatisfied that the resulting manuscript was sufficiently self-contained, so I reverted such changes. Instead, I have made following edits: both at the end of the Introduction and towards the beginning of the Results section, I more clearly spell-out a road map of the analyses to follow. This includes highlighting the primacy of the network-based analyses, but also their dependence on establishing network-process relationships which follow closely from previous work. I have also interwoven these sentiments throughout the Discussion while further discussing the importance of the functional foundation which sets the present manuscript apart from other contemporary studies of network neuroscience.

From the Introduction:

“Here, two datasets that have demonstrated macroscale gradients of cognitive control in the PFC^26,27^ are examined to investigate dynamics in the broader FPCN. […] Collectively, these analyses elucidate the integrative organization of the FPCN whose insights may be useful to understand other transmodal networks.”

From the Results:

“The analyses that follow proceed in two parts. First, areas within the FPCN are functionally mapped and related to behavior by contrasting activation and performance across different task conditions. […] These dynamics are then related to individual differences in trait-level cognitive ability and amenability to neuromodulation to establish their importance more broadly.”

I note in passing that I am not sure that I understand the comment that the results in Figure 4 follow inevitably from the way the task and brain regions’ roles are conceptualized. I can see a multitude of ways in which reaction times and activations could have associated that would not follow the depicted patterns. Given that “task-positive” areas very often rise and fall with reaction time (i.e. present behavior), it seems that it could have easily been the case that present behavior explained the rise and fall of all areas. If this continues to be a concern, I would appreciate further discussion of the matter.

As for the novelty of the effective connectivity analyses employed in the present manuscript, there are a few things that are worth emphasizing. Previously, we performed “classic” dynamic causal modeling (DCM) as a generative modeling approach to the time-series of six areas of the PFC. This approach answered questions regarding which areas, if any, might be considered the apex of the putative PFC hierarchy as operationalized by asymmetry of influence. This manuscript considers the broader question of how areas within the FPCN interact to support cognitive control. This question bridges literatures that have been narrowly focused on the organization of the PFC to literatures that have considered the network organization of cortex more globally but without fine-grained functional specificity. Classic DCM is computationally quite demanding and quickly becomes computationally infeasible when more than a handful of areas are considered. Hence, to move towards network-level descriptions, a different approach is needed. Here, I performed spectral DCM as generative modeling approach to the frequency cross-spectra of the FPCN. Despite shared nomenclature, classic and spectral DCM differ substantially in both algorithm and scope (c.f. Friston et al., 2014). Given the stationarity of the spectral DCM approach, it was then complemented by psycho-physiological interaction (PPI) analyses to detail non-stationary aspects of effective connectivity, which is another method algorithmically distinct from the previous work. Thus, the connectivity-based analyses are new analyses that follow in the spirit of recent trends to apply new, but often related analytic approaches to existing datasets (e.g. HCP, Midnight Scan Club) in order to answer novel questions. Collectively, these approaches revealed unique insights into integrative FPCN dynamics underpinning cognitive control that could not be explored by our previous approach, and provide a macroscale picture that could not be properly appreciated within our previous approach. I have added text to the Materials and methods to highlight these points more clearly.

From the Materials and methods:

“Time-invariant effective connectivity was estimated using spectral dynamic causal modeling (spDCM) implemented in SPM12 and DCM12.5. spDCM inverts a biophysically plausible generative model of the fMRI cross-spectral density^32–34^. […] An added benefit of its computational tractability is that the present approach can also scale up to answering questions about the interactions of the FPCN and other networks, which will facilitate future work.”

Finally, I would like to close with the broader point that novelty often comes from a unique combination of factors rather than any one factor alone. Classic, task-based approaches can offer strong functional inference, but such studies can often suffer from being narrow either in the processes examined, brain areas detailed, or both. Contemporary, network-based approaches consider networks in their entirety, but can be difficult to integrate into cognitive and neuroscientific theory without a firm functional foundation. I see the present study as a bridge that incorporates the best of both of these approaches that I hope serves as a paradigmatic way forward to provide mechanistic insights that are grounded in fine-grained function. Collectively then, I hope you will agree that the combination of the functional foundation, which has been expanded here relative to previous works, and new connectivity-based analyses offers sufficient novelty when considered jointly.

2) While the author frames the cortical subdivisions observed here as subcomponents of the Fronto-parietal control network, it is not entirely convincing that this is an accurate description of these areas. The "temporal control" areas, in localizing to angular gurus, far anterior middle frontal gyrus, and anterior temporal cortex, could arguably be better described as being within the Default network. The "Sensory-motor control" regions, in localizing to regions such as SPL and SFS, could be better described as being within the Dorsal Attention network. Certainly, the described differential functions of these regions fit well with known functions of these two networks. This could be adjudicated by a) showing medial surface views of Figure 2 (helpful especially for determining whether Default regions are involved), and b) computing an overlap metric of each set of regions (or activation map) with a publicly available network map, such as that described in (Yeo et al., 2011). If these regions do overlap better with non-FP networks, this doesn't necessarily invalidate any of the paper's findings, but it does change their interpretation somewhat: instead of reasoning that a central region of FP network regulates peripheral FP regions, we might conclude that FP regulates other networks, in a fashion similar to that outlined in (Gratton et al., 2018).

I agree with the editors/reviewers that making more direct contact with publicly available network maps would be useful to both better situate the networks observed here, and also offer functional insights into well-established networks whose precise functions remain unclear. I have computed the overlap between the networks mapped by the task contrasts and the Yeo et al., 2011 atlas as recommended. Below I depict the overlap of the networks mapped by the task with relevant networks described by Yeo et al., 2011 in their 17-network parcellation. Note that the “FPN”, nomenclature is in keeping with Dixon et al., 2018 and Murphy et al., 2020, which differs from Yeo’s nomenclature. This analysis indicates that the temporal control network (red) aligns most clearly with “FPN A”. This network has been recently hypothesized to be a bridge between the FPCN and DMN (Dixon et al., 2018; Murphy et al., 2020) which accords well with the idea of it being on the internal end of the FPCN axis. The sensory-motor control (blue) networks aligns most closely with “FPN B” and also showing some semblance to the “DAN”. This accords well with the sensory-control network being on the external end of the FPCN axis, and the recent hypothesis that “FPN B” is a bridge between FPCN and DAN (Dixon et al., 2018; Murphy et al., 2020). Finally, the contextual control network aligns most closely with “FPN B”, but also overlaps with “FPN A” suggesting the integration of these two networks for contextual control. Collectively, these data re-affirm the idea that the task contrasts map networks involved in cognitive control as designed. Moreover, I believe that these data serve to “ground” these connectivity-based networks more concretely in function, adding insights into the roles of these networks that cannot be established by connectivity-patterns alone. I have added a depiction of these overlaps to Figure 2 and added text to the Materials and methods, Results and Discussion accordingly.

From Materials and methods:

“For each activation contrast, the overlap with the Yeo et al.^11^ 17-network parcellation was examined. For each of the 17 networks, the percentage of significant voxels for a given activation contrast that overlapped with the network was tabulated.”

From Results:

“Previous work has suggested that the FPCN can be fractionated into sub-networks^10–13^. In particular, it has been proposed that an FPCN network A acts as an intermediary between the FPCN and internally-oriented “default-mode” network (DMN). […] Furthermore, these observations offer more precise functional descriptions of previously fractionated sub-networks.”

From Discussion:

“The position of the FPCN with respect to macroscale gradients of cortex is intermediary between canonical circuits involved in modality-specific processing, and the default-mode network involved in internal mentation^21,44^[…] Collectively, these data serve to functionally ground the sub-networks that have been identified through co-activations.”

3) While the author has framed these results in the context of a macroscale gradient in frontal (and parietal) cortex, there is no strong evidence for this. Three sets of regions with different functions does not necessarily indicate a gradient; it could indicate three discrete networks. To make strong claims about finding a gradient, the author would need to test a continuous line of ROIs from one of the current seeds to the next, and show that brain function varies continuously with position. And ideally this would need to be done in individual subjects, to avoid the blurring that comes from averaging across subjects with variable spatial organization of FP (as detailed in Braga and Buckner, 2017). When such tests have previously been done (in resting-state data, at least), they indicate the existence of strong borders between discrete areas in lateral frontal cortex, not continuous gradients (see e.g. Wig et al., 2014, especially Figure 7). It would not be completely necessary to perform such a test, because this concern doesn't invalidate any of the findings here. However, absent more comprehensive tests, the gradient interpretation might be simply only one of several frameworks that should be discussed. Towards that end, the author could make better contact with previous works that have subdivided the brain into discrete sub-networks (rather than framing it as a continuous gradient) (Abou Elseoud et al., 2011; Doucet et al., 2011; Meunier et al., 2009), and especially those that have characterized how FP subdivisions connect strongly with other networks. (Gordon et al., 2018) and particularly (Dixon et al., 2018) seem highly relevant here, and they fit very nicely with the present characterization of the central FP network providing separate top-down integrative control of internal and external stimuli.

The editors/reviewers raise an interesting question regarding the resolution of so-called “macroscale gradients.” I have added text to be clear about what I mean by “gradient” so that it is not mistaken for the idea of being “continuous” at some fine spatial scale. By “gradient”, I take the weak position that there are spatial axes of functional change along the cortex. The spatial resolution of those axes may be areal or something finer (e.g. columnar). The data as analyzed are consistent with an areal scale, but remain silent regarding whether or not finer scales exist. Given that any form of averaging/interpolation can cause discrete functions to appear continuous, I do not think that analyses on the present dataset would be convincing regarding finer scales when considering the voxel sizes acquired, preprocessing performed, and SNR of the data absent averaging/interpolation. So, I believe that an appropriate answer to this concern would require a dataset more specifically designed to address such questions. I have added a footnote in the Introduction to clarify what is meant by “gradient” in the present manuscript. I have also added text to the Discussion regarding the limitations of the present dataset to uncover the resolution of the “gradient.” Finally, I have included discussion of the suggested references along with a recent preprint that examined gradients in the non-human primate at a spatial scale finer than can be obtained with fMRI and found support for the areal level of resolution. The present data are not inconsistent with these findings.

From the Introduction:

“A “gradient” refers to a change in a property along an axis. With respect to cortex, a functional gradient means that functions vary in a roughly monotonic fashion with cortical distance along a spatial axis. The resolution of the gradient, or in other words, the steps along the axes, may be discrete and areal, or approach continuity. The present study remains silent regarding the resolution of these gradients, and takes the weak position that functions change along cortical axes at some unknown rate/resolution.”

From the Discussion:

“One question that remains open is the spatial resolution of macroscale cortical gradients. […] Because of the processing and spatial resolution of the data here, the present dataset is unable to speak to finer gradations that may exist below the areal level, and is not inconsistent with the idea that the resolution of macroscale gradients is the areal level.”

4) The results in this manuscript leave open the question of how they can be reconciled with strong evidence that multitudes of tasks co-activate highly specific regions within the fronto-parietal cortices that largely overlap with regions of the intermediate zone (e.g. Shine et al., 2016; Fedorenko et al., 2013; Assem et al., 2020). Importantly, a recent study using hundreds of subjects and improved brain alignment methods (Assem et al., 2020) showed that even a simple 0-back task (similar to the baseline task used in this study) co-activates both caudal and rostral fronto-parietal regions. Is it possible the current task contrasts are missing out on regions with signals of comparable strengths? For example, in the sensory-motor contrast (Dual + baseline > Planning + switching), could the baseline task be also engaging broader frontal and parietal regions though more weakly than the other tasks? This would suggest that the intermediate zone has relative rather than absolute concrete/abstract gradient preferences.

I believe that this is a point about the specificity observed in the data with regard to areal-process associations/dissociations relative to other studies. If I am mistaken in that interpretation, I would welcome further clarification and discussion.

It is not clear to me that the observation of activation across the FPCN when contrasting 0-back with fixation (i.e. Assem et al., 2020) has strict bearing upon the patterns observed here, and I am not sure that sample sizes and processing methods are operational in this regard. Rather, it is notable that BOLD fMRI is a unitless measure that affords only relative claims. Given the limitations of BOLD, well-matched comparisons are key for dissociating the roles of functionally related areas. For example, one can observe activation in the FFA when comparing scrambled faces to fixation. One could potentially interpret such a result as an indication that the FFA is not preferentially involved in face processing, but rather is involved in visual processing more generally. Yet, adding faces as a third comparison would reveal that the FFA responds more strongly to faces than to scrambled faces, suggesting a role in visual processing, as well as a more selective role in face processing (c.f. Kanwisher et al., 1997 among countless other works). This underscores the importance of well-controlled comparisons to dissociate functional roles of related areas (along with the need for causal methods to validate correlational patterns). Therefore, that some or all of the areas detailed here are activated for 0-back relative to fixation does not undermine more specialized control functions of some of the areas therein, just as observation FFA activation during non-face tasks does not undermine its role in face processing. Hence, I see no necessary inconsistency with the cited studies and the data reported here.

This issue speaks to contentions between accounts that regard the FPCN as functionally homogenous (or nearly so) whose organization is unclear, and accounts that regard the FPCN as functionally heterogeneous and organized along particular dimensions. I have spoken to this debate elsewhere (e.g. Badre and Nee, 2018) and I am not sure that this manuscript is the appropriate outlet to repeat/expand upon such arguments. Although the present data certainly fall on one-side of this debate, and a particularly interested reader could closely inspect the radar plots in Figure 3 to determine how much each area responds to the baseline condition alone, it is not the focus of the present study and I worry that discussing the matter would only further divert focus from the essence of the study (as per Essential revision 1). Therefore, I have opted to set this debate aside for the moment, but I would be willing to add to the Discussion if the editors feel it is essential.

5) An important methodological concern is that all the analysis used data with excessive unconstrained volumetric smoothing (8 mm) mixing signals from functionally distinct regions (see Coalson et al., 2018). Finding replicable results across two independent samples analyzed using the same non-optimal methods does not rule out the possibility of falling in the same pitfalls in both datasets. For example, while the manuscript claims to examine gradients within the fronto-parietal network, the temporal control contrast might be mainly highlighting DMN activations and connectivity dynamics. This also limits the interpretability of the connectivity indices and their relation to individual differences (Bijsterbosch et al., 2018, 2019). Ideally, the author is encouraged to repeat the task and connectivity analyses with minimal smoothing (2-4 mm). Further, why are all the analyses limited to the left hemisphere? If it is because the task effects are stronger on the left, this does not exclude decent effects on the right hemisphere which could provide a strong validation for the task and connectivity results.

After additional discussion with the editors and reviewer, I have repeated the analyses using 4 mm volumetric smoothing. In particular, follow-up discussion revealed concern that the integrative role of mid-positioned areas may have arisen from signal mixing caused by an excessively large smoothing kernel. We have agreed that 4 mm volumetric smoothing would sufficiently dampen such potential confounds and all analyses have been repeated with this processing choice.

As now reported, the observed results do not appear to be artifacts of a large smoothing kernel. Activation gradients (Figure 2—figure supplement 1), activation profiles (Figure 3—figure supplement 1), brain-behavior relationships (Figure 4—figure supplement 1), lobular source-target relationships (Figure 5—figure supplement 1), static network interactions (Figure 6—figure supplement 2), and dynamic network interactions (Figure 7—figure supplement 3) remain unchanged. Integration dynamics also continue to explain individual differences in susceptibility to TMS (Figure 9—figure supplement 1). The lone analysis that did not replicate in full was the relationship of integration dynamics to individual differences in cognitive ability. Previously, I had observed a weak negative relationship between cognitive ability and static integration (r = -0.22), and a weak positive relationship between cognitive ability and dynamic integration (r = 0.23). These integration dynamics could be jointly used to significantly predict cognitive ability (r = 0.32). In the re-analysis, the weak positive relationship between cognitive ability and dynamic integration remained intact (r = 0.26), but I no longer observed a relationship between cognitive ability and static integration (r = – 0.04). As a result, the joint use of these variables no longer predicted cognitive ability (r = 0.03).

I am unaware of a straightforward mapping of smoothing kernel choice to the change in this result, especially in light of the many results that did replicate. I performed a variety of validation checks which were unremarkable. The only matter of note was a significant reduction in the explained variance of signals summarizing each individual/area. To unpack this: each area for each individual was summarized into a single time-series using the first eigenvariate of the time-series’ of the voxels comprising the area. With reduced smoothing, the first eigenvariate explained significantly less variance of each area’s time-series, which is unsurprising given that smoothing would be expected to reduce each area’s heterogeneity. One possibility is that this reduction in explained variance inflated noise in network estimates. I am not convinced of this possibility – I checked the test-retest reliability of the spectral DCM parameters (from which static integration is computed) for sample 1 (which performed 2 sessions) and they were similar with 8 mm and 4 mm smoothing (>0.6 on average in both cases). The group-level replicability of spectral DCM parameters remained excellent with 4 mm smoothing (>0.95). This suggests that a simple explanation regarding increased noise with less smoothing is insufficient.

Given the above, I am left with two interpretations that I do not think can be readily resolved at the moment. First, it is possible that with reduced smoothing, the resultant signals used to summarize each individual/area were less representative of an individual’s/area’s population activity, thereby reducing the ability to explain individual trait-variability. That is, even if stability remained unchanged, the computed measures themselves may no longer be adequately capturing the individual variability that is relevant for behavior. Behavior arises from population-level dynamics such that considering more of the relevant population-level signals should facilitate better explanations of behavior. Hence, while less smoothing makes the signals more focal, which is good for dissociating nearby areas, it may be that pooling across nearby areas makes the signals more representative of the population-level dynamics that explain behavior. Towards this possibility, prediction of individual susceptibility to TMS was reduced in magnitude, though still significant, with less smoothing. It is possible that this reduction reflects regression towards the mean. That is, it is often the case that effect sizes that are conditioned on significance are smaller when investigated with alternative analysis strategies/samples. Mitigating this concern, entirely new analyses using activation features to predict cognitive ability also showed poorer prediction accuracy with less smoothing. These analyses are unlikely to reflect regression to the mean since these analyses were performed simultaneous on both preprocessing pipelines.

I think a systematic investigation of this matter could be performed with a careful and complete parcellation of the FPCN that takes advantage of the tasks’ detailed functional profiling. Consider the possibility that all areas within the FPCN contribute to cognitive ability, but only a subset are summarized for analyses, as is the case here. In such a case, more smoothing may allow more of the relevant areas/signals to be represented in the analysis. An alternative approach would be to analyze all areas of the FPCN. If this could be done, then representation becomes a non-issue, and less smoothing would help to preserve distinctions among areas that may be useful for understanding individual variability. This will necessarily be a more difficult analysis – it will be non-trivial to appropriate align all relevant functional areas across participants especially in light of heterogeneity in the FPCN (e.g. Braga and Bucker, 2017; Gordon et al., 2017; Gratton et al., 2018). The rich task data here, along with connectivity profiling, may help to facilitate such alignments. However, appropriate investigation into this matter is likely to be a study in its own right, and is left as a future direction.

The second possibility is that the original result was a false positive and/or this non-replication indicates instability of effect size estimates at the present sample size (i.e. insufficient power). Given the sample size, the originally observed prediction effect was just within reasonable power reach (i.e. ~80% power). The failure to replicate this effect could mean that the true effect size is smaller than originally observed, if present at all. A smaller such effect would not be detectable in the present dataset, and the observation of such an effect could be considered spurious. I do not believe that the present dataset can satisfactorily speak to this possibility, and replication with a larger sample would be necessary. For the moment, collection of such a dataset is not feasible, and it is not clear how to perform analogous analyses in large open datasets.

Given these issues, I have added discussion of these matters and the warning that this analysis should be interpreted with caution. I also wrestled with the idea of removing the analysis from the manuscript entirely, but I thought it better to document and discuss appropriately rather than to “file-drawer” the analysis.

From the Results:

“The same analyses were repeated with reduced smoothing. Although dynamic integration was again positively associated with cognitive ability, with reduced smoothing, no relationship was observed among cognitive ability and static integration. As a result, integration measures could not be used to predict cognitive ability (r = 0.03; p = 0.44; Figure 8—figure supplement 2). On the other hand, activation based measures remained significantly predictive (r = 0.31, p = 0.04; Figure 8—figure supplement 3) and significantly improved prediction when added to models containing integration measures alone (r = 0.32, p = 0.03; nested model comparison using ordinary regression: F(4,42) 3.05, p = 0.04). I return to the implications of the non-replication with reduced smoothing in the Discussion.”

From the Discussion:

“Third, multiple steps have been taken to ensure the replicability of the findings here including replicating results across independent samples, and repeating the analyses with different preprocessing choices. […] Whether the lack of replication of FPCN integration measures predicting cognitive ability with reduced smoothing indicates that the original result/analysis was a false positive/had insufficient power or whether this is a reflection of a true result varying as a function of preprocessing choices requires future investigation. For the time-being, this result should be taken with some measure of caution.”

Although I have focused a large amount of this response on the one result that did not satisfactorily replicate with the change in smoothing kernel, I would like to reiterate that all other analyses did, in fact, replicate. I believe this should quell notions that the results are, by-and-large, an artifact of a somewhat arbitrary preprocessing choice.

Finally, with regard to hemisphere, it is common for tasks in the abstraction/hierarchy literature to be more prominently left lateralized. I have commented on this in the past when meta-analytically gathering activation foci from related works (c.f. Nee et al., 2014; Badre and Nee, 2018). Although strong bilateral activations were observed in the present study (maps are available on OSF as indicated in the manuscript https://osf.io/938dx/, as well as in NeuroVault https://neurovault.org/collections/6054/), the stimulus domain factor only showed differentiation in the left hemisphere (i.e. no area showed a preference for the verbal conditions in the right hemisphere). Therefore, the choice was made previously, and carried through here, to focus on the left hemisphere. This leaves open characterization of hemispheric differences, which I hope to follow-up on, but are outside the scope of the present manuscript. I have included additional discussion detailing limitations of the present work given the focus on the left hemisphere alone.

From the Discussion:

“Fourth, the analyses presented here have focused exclusively on the left hemisphere. This is in keeping with prior work with these data^26,27^ and is consistent with the preponderance of work in the domain of abstraction and hierarchical control showing more marked recruitment of the left hemisphere^4,38,78–84^. The reason for this left lateralization is unclear, but may relate to the presumed development of internalized control structures via interactions with the language system^85,86^. However, some control processes, such as those responsible for the inhibition of motor responses tend to be more right lateralized^87,88^. Future work would do well to compare and contrast network integration of the left and right FPCN.”

6) To improve the interpretability of the effective connectivity results, it would be helpful to provide more details on the methodological and biological assumptions behind using positive and negative effective connectivity indices to suggest that they reflect excitation and inhibition, respectively, as well as the limitations related to fMRI effective connectivity analyses. Further, negative correlations between static connectivity and general cognitive abilities are in contrast with numerous reports showing that measures of resting-state functional connectivity across the fronto-parietal network positively correlate with general cognitive abilities (e.g. Dubois et al., 2018). Could the author comment on this difference?Also, the author interprets the negative between-network connectivity as reflecting functional segregation, and positive connectivity as reflecting integration. This is plausible but certainly not the only possible interpretation of such a pattern. For instance, increases in between-node connectivity with greater control demand are often interpreted as reflecting modulation/biasing of one region by the other, which seems conceptually distinct from "integration", at least the way the author conceptualizes integration (i.e., the mid-lateral PFC joining up information from posterior and anterior regions). I wonder whether there might be stronger tests of the claim that this node integrates information processed in the other nodes, perhaps by using an information-based analytic approach (like MVPA/RSA)?

I have added additional discussion regarding the assumptions of the utilized effective connectivity approaches, and how those assumptions impact interpretation of the results. As is always the case with fMRI, the neurophysiological underpinnings remain somewhat unclear, and conclusions need to be measured accordingly.

From the Discussion:

“Several limitations should be considered to properly contextualize the findings reported here. […] Hence, these observations should be taken as a starting point from which more complex and biologically plausible networks can be expanded.”

With regard to the relationship between static connectivity and cognitive ability, I am grateful for the reference provided, which I have added to the manuscript. There is a broader discussion to be had regarding how to reconcile (A) that modular brain organizations are related to better cognition, (B) that strong within-network connectivity is related to better cognition, and (C) that classically defined networks are composed of sub-networks. This leads to the question of whether modularity at the sub-network level is beneficial or harmful to cognition, which appears to be an open question. I suggest that static integration reflects less modularity at the sub-network level, and commensurately poorer cognitive ability, but this warrants follow-up with a graph theoretic approach, ideally on a larger dataset. I have added text to the discussion regarding this interesting future direction.

From the Discussion:

“Previous work has indicated the presence of an integrative core of regions important for multiple tasks^14–17,25,53–56^. […] Our data situate the areas associated with such integration in an intermediary zone of the FPCN.

In contrast to the integrative dynamics of intermediary areas, sensory-motor proximal and distal areas acted in a segregative fashion. Segregation is likely to be important to select relevant information while suppressing irrelevant information. Consistent with the data here, Shine^57^ posited that rostral areas of the PFC are involved in segregation while mid-lateral areas are involved in integration. He theorized that segregation is mediated by cholinergic modulations while integration is mediated by noradrenergic modulations. Further research into the influences of distinct PFC-PPC networks on neuromodulatory mechanisms would be fruitful to elucidate such effects.

The data demonstrated both static and dynamic forms of integration. Areas involved in contextual control tended to excite other control networks, providing a potential substrate for integration through binding. Moreover, demands on contextual control increased inter-network communication producing a dynamic form of integration. Both static and dynamic integration were related to individual differences in higher-level cognitive ability. Interestingly, the different forms of integration tended to be associated with opposite effects. On the one hand, increased static integration was associated with decreased higher-level cognitive ability. It has been posited that an appropriate balance of segregation and integration into distinct networks or modules is important to optimize brain efficiency^58^. In particular, it has been observed that more segregation between networks, and integration within networks (i.e. modularity) at rest is associated with memory capacity^59^. Therefore, integration across networks in a stationary manner may be sub-optimal for cognition. On the other hand, increased dynamic integration was associated with increased higher-level cognitive ability. These data are consistent with previous studies that have shown that integrative reconfigurations of brain networks from rest to task are related to improved performance on complex tasks^60–62^. These data suggest that the brain’s ability to integrate “on-demand” is beneficial to cognitive processing.”

Finally, the editors/reviewers raise an intriguing idea that an information-based approach might be able to adjudicate among mechanisms providing top-down biasing on the one hand and integration on the other. I think this is a pressing question that has recently been elegantly discussed by Badre and colleagues (https://psyarxiv.com/asdq6/). The appropriately conceived analysis/dataset to address this question would have widespread impact for the study of cognitive control and brain function. I tend to think that addressing this question is a study in its own right that would in the very least require a different analysis pipeline (e.g. no smoothing or spatial normalization to preserve insofar as possible the multi-variate signals), but would likely also require an innovative analytic approach, and a dataset more specifically designed to address the question (e.g. Badre and colleagues suggest repetition suppression designs might be necessary given limitations of multi-variate signals in the PFC as measured by fMRI). Therefore, although I would very much like to pursue an answer to this question, I feel as though it is outside the scope of the present manuscript. In lieu of addressing this question directly, I have added discussion of these matters. First, I recognize a modulation-based interpretation, offer why I favor an integration view, but also admit that new data/analysis will be needed to resolve this matter:

From the Discussion:

“Although I have suggested an integrative role of areas positioned in the middle of the FPCN, other accounts are plausible. […] Resolving this matter would likely require additional data that can identify the distinct representations of different areas of the FPCN and how those representations are modified by network dynamics.”

7) It would be problematic at brain-behavior individual differences using a small sample size like the one used to probe connectivity-TMS effects due to their unreliability. If the author insists on keeping this section, it will be useful to examine any shared variance between general cognitive abilities and the observed TMS behavioral effects.

I agree that a larger sample size would be preferable for exploring the relationship between connectivity and TMS. That said, I hope the editors/reviewers will indulge me a philosophical point so as to speak to recent trends that I think are unfairly trivializing smaller n studies: the utility of small, carefully collected samples should not be dismissed as evidenced by classic work on lesion patients, extensive insights gleaned from the neurophysiology of pairs of monkeys, and the many recent impactful findings garnered from the Midnight Scan Club dataset. While it would be desirable to perform individual difference analyses of TMS-connectivity relationships on datasets the size of HCP or ABCD, no such dataset exists. Given the non-triviality and cost of functionally defining multiple TMS targets at the individual level, and then assessing TMS effects in multiple follow-up sessions, one would probably want good cause prior to pursuing large n investments of such designs. Consider that large datasets such as HCP would never have been collected had it not been for studies with sample sizes on the order of the one explored here paving the way with both promises and pitfalls. Hence, analyses such as these are essential for showing sufficient promise to warrant the investment needed to collect larger samples with those larger samples then providing more definitive conclusions. In other words, we need small data before we can get big data, which I feel is an important point for proponents of big data to recognize and support. On the other hand, conclusions based on small data should be appropriately measured.

With that said, I am in full agreement that the sample sizes limit strong conclusions. That the observed relationships continued to be significant even after re-processing the data speaks to the idea that there is some reliable relationship here. However, replication with an independent, larger sample is certainly needed. I have added discussion regarding the tentative nature of the observed results given these issues.

From the Discussion:

“Second, conclusions regarding the relationships among individual differences in FPCN network integration and cognitive ability/susceptibility to neuromodulation should be tempered by the sample sizes studied. […] However, replication with larger samples is needed to draw firm conclusions regarding the relationships among FPCN network integration, cognitive ability, and susceptibility to neuromodulation.”

I have also examined and detailed other related variables as recommended. First, there was a non-significant negative correlation between cognitive ability and TMS susceptibility (r = -0.33). To examine whether integration measures predicted TMS susceptibility over-and-above cognitive ability, cognitive ability was regressed out of measures of static and dynamic integration. The resultant predictors still significantly predicted TMS susceptibility in the data as originally processed (r = 0.46, p = 0.02), and in the data with reduced smoothing (r = 0.41, p = 0.04). Second, as detailed below, activations could be used to predict cognitive ability. Interestingly, activations alone could not be used to predict TMS susceptibility (correlations among predicted and observed TMS effects all <0). Finally, integration measures predicted TMS susceptibility after regressing out both cognitive ability and activations in the data as originally processed (r = 0.47, p = 0.02), and in the data with reduced smoothing (r = 0.37, p = 0.045). [Note, below I use a model comparison approach rather than controlling for variables as I do here. I control for variables here because the small sample size precludes adding too many predictors, thereby making a model comparison approach inappropriate, as far as I can tell.] Although sample sizes preclude strong conclusions, these data indicate the utility of connectivity-based measures to predict TMS effects. I have added these details to the manuscript.

From the Results:

“Next, additional relationships among TMS susceptibility, cognitive ability, and activation were examined. […] Hence, these data suggest that network integration is useful for predicting susceptibility to neuromodulation over-and-above cognitive ability and control-related activations.”

[Editors' note: further revisions were suggested prior to acceptance, as described below.]

1) In the Discussion section "Beyond the PFC: Towards a Network View of Cognitive Control" the author gives the impression that the majority of studies on cognitive control have solely focused on the PFC. This needs to be rebalanced to reflect that a distributed whole brain network view of cognitive control has long been argued (to name a few: Cole and Schnider, 2007; Power and Petersen, 2013; Fedorenko et al., 2013; Warren et al., 2014; Duncan et al., 2010 and 2020).

I apologize for misrepresenting the literature. It was my intention to address the literature on hierarchical cognitive control that has been disproportionately dominated by examinations of the lateral PFC. I have edited this section accordingly, adding in the recommend references along with others, while being careful to more clearly indicate the more specific focus that I am attempting to expand with this study:

From the Discussion

“A substantial body of work has focused on the functional organization of the lateral PFC and interactions therein that support cognitive control. Despite long-standing recognition that cognitive control is supported by areas distributed across the frontal and parietal lobes^9,11,15,17,38–42^, much work, particularly in the domain of hierarchical cognitive control, has centered narrowly on how processing varies along the rostral-caudal axis of the PFC^4,22,23,26,27,43,44^. Although a number of insights have been gained by focusing on the PFC, such a narrow focus ignores the broader networks in which the PFC participates. Choi^28^ recently demonstrated that the same rostral-caudal organization for control observed in the PFC is reflected in the PPC consistent with the idea that the PFC and PPC are comprised of ordered networks for control. The data here replicate those findings while also linking PFC-PPC activations with behavior along distinct timescales. Hence, it appears to the case that many functions that have been attributed to the PFC are also present in the PPC. This necessitates expanding the study of cognitive control beyond the PFC to network and brain-wide levels.”

2) Some of the supplementary figures were missing (Figure 2—figure supplement 2, Figure 6—figure supplement 2, Figure 7—figure supplement 3), not sure if this was a problem with the submission system. There were also a few typos.

My apologies for these oversights. I have edited the typos and uploaded the missing figures.